# Single-cell dynamics of pannexin-1-facilitated programmed ATP loss during apoptosis

Hiromi Imamura[1][†]*, Shuichiro Sakamoto[1][†], Tomoki Yoshida[1], Yusuke Matsui[2], Silvia Penuela[3], Dale W Laird[3], Shin Mizukami[4], Kazuya Kikuchi[2], Akira Kakizuka[1]

[1]Graduate School of Biostudies, Kyoto University, Kyoto, Japan; [2]Graduate School of Engineering, Osaka University, Suita, Japan; [3]Department of Anatomy and Cell Biology, Schulich School of Medicine and Dentistry, University of Western Ontario, London, Canada; [4]Institute of Multidisciplinary Research for Advanced Materials, Tohoku University, Sendai, Japan

**Abstract** ATP is essential for all living cells. However, how dead cells lose ATP has not been well investigated. In this study, we developed new FRET biosensors for dual imaging of intracellular ATP level and caspase-3 activity in single apoptotic cultured human cells. We show that the cytosolic ATP level starts to decrease immediately after the activation of caspase-3, and this process is completed typically within 2 hr. The ATP decrease was facilitated by caspase-dependent cleavage of the plasma membrane channel pannexin-1, indicating that the intracellular decrease of the apoptotic cell is a 'programmed' process. Apoptotic cells deficient of pannexin-1 sustained the ability to produce ATP through glycolysis and to consume ATP, and did not stop wasting glucose much longer period than normal apoptotic cells. Thus, the pannexin-1 plays a role in arresting the metabolic activity of dead apoptotic cells, most likely through facilitating the loss of intracellular ATP.

*For correspondence:
imamura.hiromi.7a@kyoto-u.ac.jp

[†]These authors contributed equally to this work

Competing interests: The authors declare that no competing interests exist.

## Introduction

Living cells require energy that is provided by the principal intracellular energy carrier, adenosine-tri-phosphate (ATP). Free energy from ATP that is released upon hydrolysis is utilized in various vital processes including the generation and maintenance of the plasma membrane potential (*Kaplan, 2002*), remodeling of chromatin (*Vignali et al., 2000*), locomotion of molecular motors (*Vale, 2003*), protein degradation (*Ciechanover, 1994*), and metabolic reactions (*Voet and Voet, 2010*). Therefore, living cells must maintain high concentrations of intracellular ATP by continuously regenerating ATP from its hydrolysis products, adenosine-diphosphate (ADP) and phosphate ion, via energy metabolism that uses chemical energy stored in cellular nutrients, such as glucose. On the other hand, dead cells contain very little, or even no ATP.

Apoptosis is a form of programmed cell death, with important roles in development, tissue homeostasis, and immunity. It is characterized by distinctive morphological changes, such as membrane blebbing, nuclear condensation, and externalization of phosphatidylserine (*Elmore, 2007*). Although there are a variety of stimuli that can provoke apoptosis, these stimuli converge into the activation of a proteolytic cascade of cysteine proteases, called caspases. Because cleavage of specific target proteins by activated effector caspases triggers apoptotic events, including the characteristic morphological changes, apoptotic cell death is a systematically and genetically determined, or 'programmed', process. In contrast, necrosis is considered to be the 'unprogrammed', cataclysmic demise of the cell.

It has been reported that cells die from necrosis rather than apoptosis when intracellular ATP is depleted prior to otherwise apoptotic stimuli (*Eguchi et al., 1997*; *Leist et al., 1997*). It has been also reported that dATP/ATP is required for the formation of cytochrome c/Apaf-1/procaspase-9 complexes (*Hu et al., 1999*; *Li et al., 1997*). Moreover, it is suggested that ATP is required for chromatin condensation of apoptotic cells (*Kass et al., 1996*). Apoptosis is, thus, considered to be an energy-demanding process, requiring intracellular ATP for the execution of the cell death program. In spite of, or perhaps in part due to the requirement of ATP for apoptosis, the intracellular ATP in apoptotic cells is ultimately depleted. Therefore, it seems likely that both maintenance and reduction of the intracellular ATP level are systematically regulated during the progression of apoptosis. Intracellular ATP levels have been conventionally analyzed with the firefly luciferin-luciferase system (*Lundin and Thore, 1975*), liquid chromatography (*Sellevold et al., 1986*), or related methods, which use lysates of a large number of cells. Because apoptosis progresses differently between cells (for example see *Goldstein et al., 2000*), even those cultured and stimulated in identical conditions, it has been quite difficult to precisely understand the dynamics of the intracellular ATP level in each dying apoptotic cell, and difficult to tell whether the ATP decrease accompanies specific apoptotic events. Furthermore, the molecular mechanism of how intracellular ATP decreases in apoptotic cells also remains to be elucidated. In addition, it is totally unclear whether the depleted intracellular ATP in an apoptotic cell benefits the dying cell itself or the surrounding, healthy cells.

In this work, we established a method for imaging both ATP concentration and caspase-3 activity in a single apoptotic cell with newly developed genetically encoded Förster resonance energy transfer (FRET)-based biosensors for ATP and caspase-3 activity. We found that the intracellular ATP level starts to decrease following the activation of caspase-3 and that the caspase-triggered opening of the plasma membrane channel pannexin-1 (PANX1) is the major cause of the decrease in intracellular ATP.

## Results

### Development of FRET biosensors for dual imaging of ATP and caspase-3 activity of apoptotic cells

In general, the progression of apoptosis varies between individual cells, even in the same cell type. It is, thus, essentially difficult to understand how the intracellular ATP level in a particular cell changes during apoptosis using conventional biochemical analyses of pooled cells, such as firefly luciferase assays. To reveal the dynamics of intracellular ATP levels during apoptosis at the single-cell level, we first used a genetically encoded FRET-based ATP biosensor, ATeam (*Imamura et al., 2009*), which is comprised of a cyan fluorescent protein (CFP; mseCFP), an $F_oF_1$-ATP synthase $\varepsilon$ subunit and yellow fluorescent protein (YFP; cp173-mVenus). Unfortunately, we found that the original ATeam (AT1.03) was cleaved into its constituent pair of separate fluorescent proteins in apoptotic cells, most probably by activated caspases (*Figure 1—figure supplement 1*). Thus, the FRET signals of the original biosensor were reduced in apoptotic cells irrespective of the ATP concentration. We replaced Asp-242 and Asp-339 of AT1.03, which we predicted were within the target sequences of the caspases, with Asn and Gly, respectively, and found that the altered ATeam was not cleaved inside apoptotic cells (*Figure 1—figure supplement 1*). We subsequently used this caspase-resistant ATeam (AT1.03CR) to study the ATP dynamics in apoptotic cells.

The dynamics of cytosolic ATP levels throughout the apoptotic process were investigated by imaging single human cervical adenocarcinoma (HeLa) cells expressing AT1.03CR. Overall, the intracellular ATP levels remained almost constant for several hours after stimulation. The cytosolic ATP levels in these cells started to decrease after a variable time interval (typically from 3 to 8 hr after apoptotic stimulation, see *Figure 1—figure supplement 2*). Once the intracellular ATP levels started to decrease, they were depleted within typically 0.5–2 hr.

The presence of a pan-caspase inhibitor zVAD-fmk almost completely blocked the cytosolic ATP decrease of anti-FAS-induced apoptotic cells (*Figure 1A*). Thus, the ATP decrease induced by apoptotic stimuli is most likely a caspase-dependent process. Next, we developed a FRET-based caspase-3 biosensor O-DEVD-FR by connecting an orange fluorescent protein mKOκ (*Tsutsui et al., 2008*) and a far-red fluorescent protein mKate2 (*Shcherbo et al., 2009*) by a Gly-Gly-Asp-Glu-Val-Asp-Gly-Thr linker containing a *bona fide* caspase-3 recognition sequence (*Figure 1—figure*

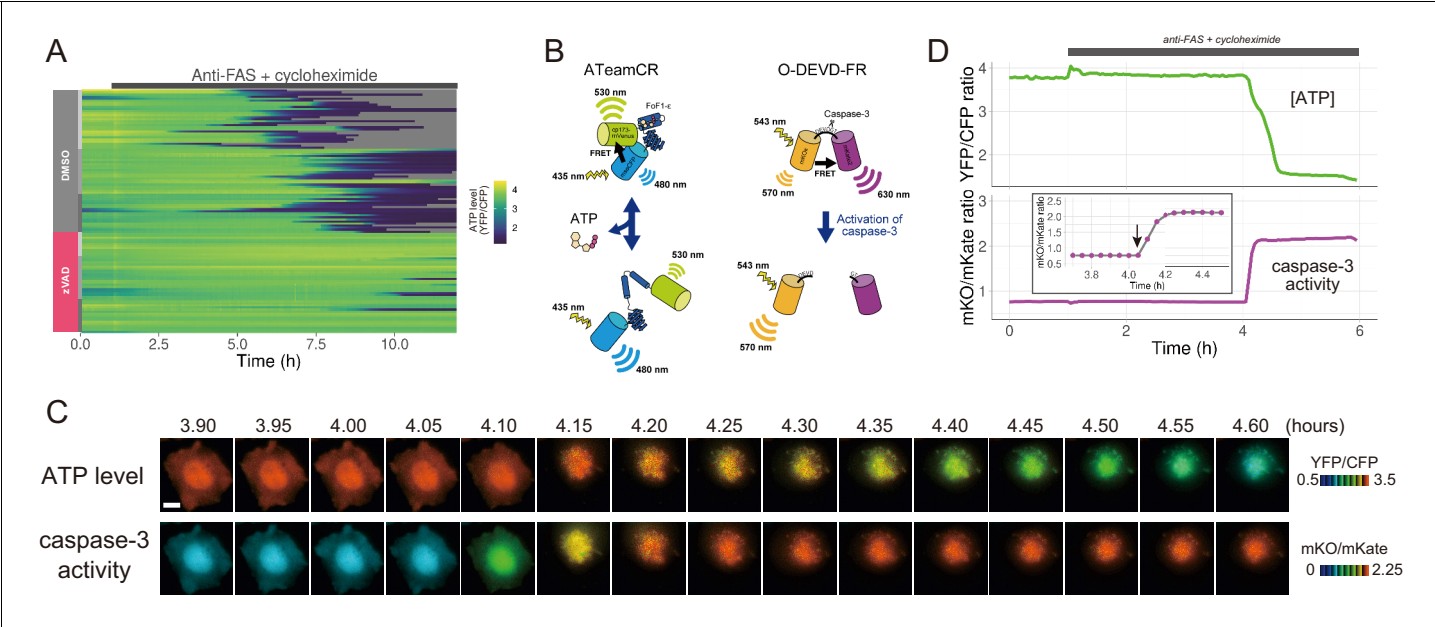

**Figure 1.** The initiation of cytosolic ATP decrease occurs almost simultaneously with that of caspase-3 activation. (**A**) Single-cell ATP dynamics after anti-FAS treatment in the absence and the presence of zVAD-fmk. Each line represents the time course of the YFP/CFP ratio of AT1.03CR from a single cell (70 [DMSO] and 49 [zVAD] cells from three biological replicates). Cells were treated with anti-FAS and cycloheximide at time = 1.0 hr. DMSO (0.1%) or pan-caspase inhibitor zVAD-fmk (20 μM) was added just before the start of imaging experiment. (**B**) Schematic drawings of AT1.03CR and O-DEVD-FR. (**C**) Time-lapse images of the ATP level and caspase-3 activity of a single apoptotic HeLa cell expressing AT1.03CR and O-DEVD-FR. Pseudocolored ratio images of AT1.03CR are shown in the upper panel, and those of O-DEVD-FR are shown in the lower panel. Bar, 10 μm. (**D**) Time courses of ATP level and caspase-3 activity of a single apoptotic HeLa cell. YFP/CFP ratio of AT1.03CR and mKO/mKate ratio of O-DEVD-FR were shown in the upper and the lower panels, respectively. Arrow in the inset indicates the onset of caspase-3 activation. Fluorescence images were captured every 3 min. The online version of this article includes the following figure supplement(s) for figure 1:

**Figure supplement 1.** AT1.03CR, a caspase-resistant mutant of the FRET-based ATP biosensor ATeam.

**Figure supplement 2.** Dynamics of cytosolic ATP levels in individual apoptotic cells.

**Figure supplement 3.** Nucleotide and amino acid sequences of O-DEVD-FR.

*supplement 3*). Once caspase-3 is activated, it cleaves the linker in O-DEVD-FR, resulting in the separation of mKOκ and mKate2, and the consequent reduction in FRET signal in apoptotic cells (*Figure 1B*). It was recently demonstrated that mKOκ-mKate2 FRET pair is compatible with CFP-YFP FRET pair because they use different spectral windows (*Watabe et al., 2020*). Thus, it is possible to use both biosensors to fluorescently image ATP level and caspase-3 activity in the same apoptotic cell (*Figure 1C,D*). Activation of caspase-3 was clearly observed as a decrease in FRET signal (an increase in mKO/mKate ratio). We defined onset of caspase-3 activation as the frame immediately preceding the first frame in which the increase in mKO/mKate ratio was first observed (see an arrow in the inset of *Figure 1D*). It was observed that intracellular ATP started to decrease after the onset of caspase-3 activation, also supporting that the ATP decrease of the apoptotic cell is a caspase-dependent process. It should be noted that any increase or decrease in fluorescence intensity due to cell morphological change was offset because we monitored the ratios of fluorescence intensities of an acceptor and a donor of the FRET biosensors.

## Single-cell dynamics of cytosolic ATP of apoptotic PANX1-knockout cells

PANX1 belongs to the innexin/pannexin superfamily and forms a heptameric pore in the plasma membrane (*Deng et al., 2020*; *Michalski et al., 2020*; *Qu et al., 2020*), functioning as a large-pore channel capable of passing small molecules (*Bao et al., 2004*; *Dahl and Muller, 2014*; *Panchin et al., 2000*; *Penuela et al., 2013*). It has been reported that apoptotic cells release ATP, AMP, and also UTP through the PANX1 channel as 'find-me' signals to attract macrophages and that the PANX1 channel is opened by caspase-3/7, which cleaves the C-terminal region of the

channel (*Chekeni et al., 2010*; *Elliott et al., 2009*; *Yamaguchi et al., 2014*). The previous cell population-based study has reported that accumulation of extracellular adenine nucleotides correlates with decreases in intracellular ATP during the apoptotic progression of Jurkat cells (*Boyd-Tressler et al., 2014*). In order to investigate the impact of PANX1 on intracellular ATP dynamics during apoptosis at a single-cell level with high temporal resolution, we utilized the dual imaging setup for ATP and caspase-3 activity to PANX1-knock out (KO) HeLa cell lines (PANX1-KO1 and PANX1-KO2), which were generated using a CRISPR-Cas9 system (*Figure 2A*). Strikingly, decreases in the intracellular ATP levels of PANX1-KO cells after caspase-3 activation were significantly slower than those of wild-type HeLa cells when apoptosis was induced by anti-FAS antibody (*Figure 2B–D*). Knockout of PANX1 apparently has no effect on the ability of the cells to undergo cell death. A marked suppression of ATP decrease by knockout of PANX1 was also observed when apoptosis was induced by staurosporine (*Figure 2E,F*). Moreover, knockout of PANX1 also suppressed the decrease in ATP during TRAIL-induced apoptosis of SW480 human colorectal adenocarcinoma cells (*Figure 2—figure supplement 1*). Thus, PANX1 is involved in the facilitation of intracellular ATP decreases during apoptosis in multiple cell types and on various apoptotic stimuli. Notably, the intracellular ATP levels of PANX1-KO cells were almost unchanged in the first 30–60 min after the activation of caspase-3, followed by a gradual ATP decline (*Figure 2C–F*, *Figure 2—figure supplement 1*). The lag in the cytosolic ATP decrease observed for PANX1-KO cells might be partially relevant to the previous observation by Zamaraeva (*Zamaraeva et al., 2005*), which suggested the enhancement of cytosolic ATP level after apoptotic stimulation. Single-cell imaging also provided unexpected observations that intracellular ATP concentration transiently and repeatedly re-elevated on the course of the gradual ATP decrease in some populations of the PANX1-KO cells (*Figure 3*). Although the mechanism for these fluctuations in the intracellular ATP concentrations is unknown at present, the fluctuations must reflect either fluctuation in the rate of regeneration of ATP from ADP, that of adenosine nucleotide synthesis through de novo/salvage pathways, or that of degradation/release of ATP, or a combination thereof.

## Effect of exogenous expression of PANX1 on single-cell dynamics of cytosolic ATP during apoptosis

Next, we exogenously expressed wild-type PANX1 in PANX1-KO HeLa cells. Cells overexpressing wild-type PANX1 precipitously lost their intracellular ATP, concomitant with the onset of caspase-3 activation, for both anti-FAS- and staurosporine-induced apoptosis (*Figure 4A–E*), further confirming that the PANX1 channel plays a major role in facilitating intracellular ATP decrease of apoptotic cells.

To examine that the intracellular ATP decrease in apoptotic cells is dependent on caspase-3 activity, we investigated the intracellular ATP dynamics of single apoptotic wild-type HeLa cells overexpressing a D376A/D379A mutant of PANX1 (PANX1-CR), in which the caspase recognition sequence close to the C-terminus of PANX1 is mutated (*Chekeni et al., 2010*). When PANX1-CR is overexpressed, most of the intrinsic wild-type PANX1 molecules are predicted to form hetero-heptamer with overexpressed PANX1-CR. The PANX1 hetero-heptamer will be expected to have much less channel activity than the wild-type PANX1 homo-heptamer as caspase-3 cannot separate the inhibitory C-terminal region from PANX1-CR inside the hetero-heptamers. In fact, it has been reported that the overexpression of PANX1-CR significantly suppresses the release of ATP from apoptotic cells (*Chekeni et al., 2010*). Accordingly, the intracellular ATP depletion was significantly protracted by the overexpression of PANX1-CR (*Figure 4F*; *Figure 4G*), clearly indicating that the cleavage of the C-terminal region of PANX1 by caspases is required for the PANX1-dependent intracellular ATP decrease of apoptotic cells.

## Single-cell dynamics of cytosolic ATP during apoptosis under an OXPHOS-dominant culture condition

It is known that cells in normal adult tissues preferentially use oxidative phosphorylation (OXPHOS) in mitochondria for the regeneration of ATP, while cells in embryonic tissues and tumors use glycolysis (*Vander Heiden et al., 2009*). In the experiments described thus far we used HeLa cells cultured in glucose-containing medium. The cells preferentially regenerate ATP by glycolysis rather than OXPHOS under these conditions. To examine the role of PANX1 in apoptosis of OXPHOS-dominant

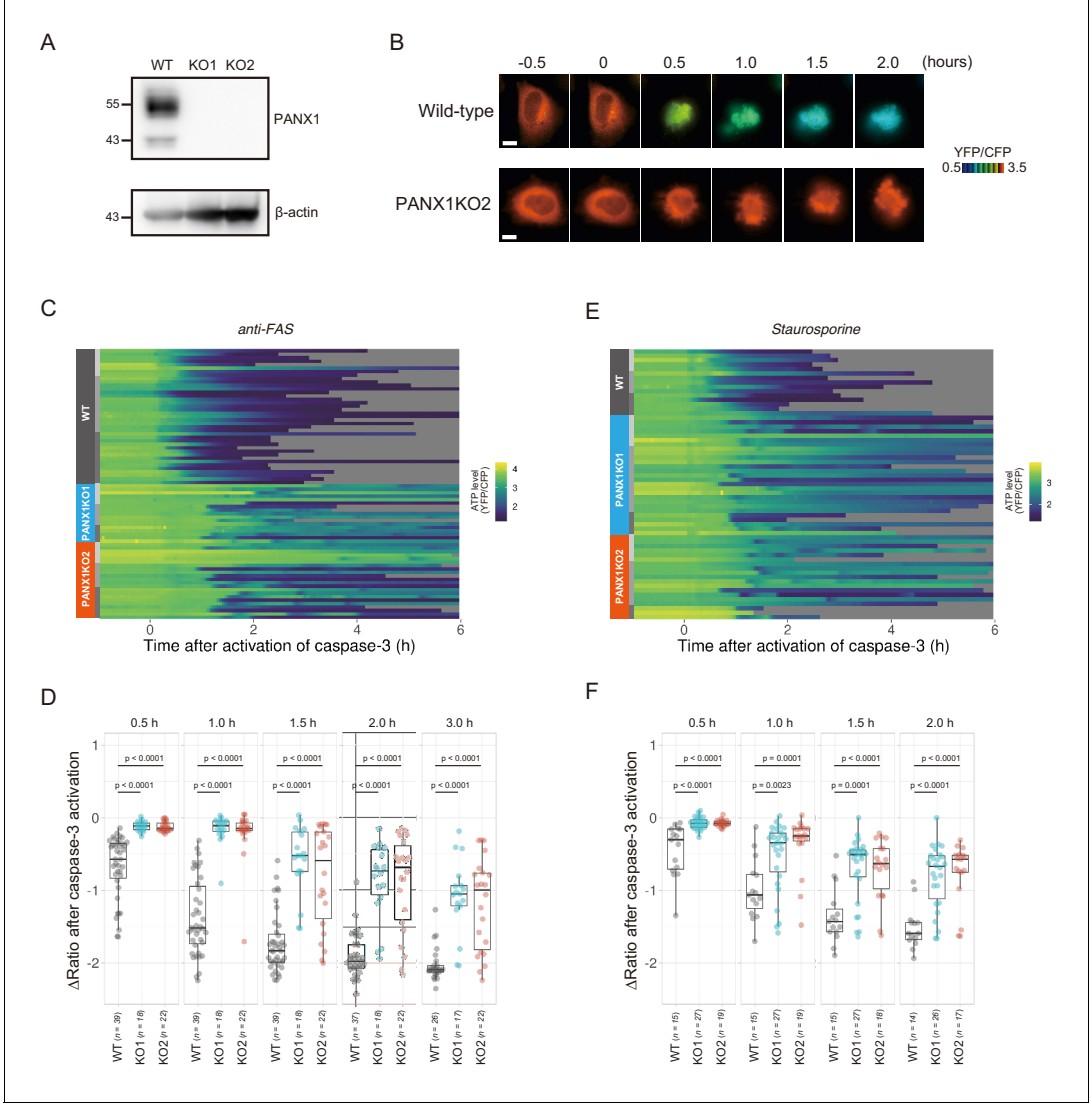

**Figure 2.** Knockout of PANX1 suppresses the cytosolic ATP decrease during apoptotic progression. (**A**) Western blot analysis of PANX1 expression of wild-type and PANX1-KO HeLa cells. (**B**) Time-lapse images of the ATP level of a wild-type cell and a PANX1-KO cell. Pseudocolored ratio images of AT1.03CR are shown. Apoptosis was induced by anti-FAS and cycloheximide. The onset of caspase-3 activation was set as time = 0. Bar, 10 μm. (**C, E**) Single cell ATP dynamics of wild-type and PANX1-KO cells during apoptosis. Apoptosis was induced by anti-FAS and cycloheximide (C; 39 [WT], 18 [PANX1-KO1] and 22 [PANX1-KO2] cells from three biological replicates), or staurosporine (E; 15 [WT], 27 [PANX1-KO1] and 19 [PANX1-KO2] cells from three biological replicates). Each line represents the time course of the YFP/CFP ratio of AT1.03CR from a single cell, and was adjusted by setting the onset of caspase-3 activation as time = 0. Traces from different replicates were labeled with bars of different shades. (**D, F**) Effect of PANX1 knockout on the decrease in cytosolic ATP levels. Changes in YFP/CFP ratios at indicated time after the onset of caspase-3 activation were calculated for each apoptotic cell. Apoptosis was induced by anti-FAS and cycloheximide (**D**), or staurosporine (**F**). Analysis of variance (ANOVA) followed by post-hoc Dunnett's test (versus wild-type).

The online version of this article includes the following figure supplement(s) for figure 2:

**Figure supplement 1.** Knockout of PANX1 suppresses the cytosolic ATP decrease of SW480 cells during apoptosis.

**Figure supplement 2.** PANX1 knockout significantly suppresses the decrease in cytosolic ATP levels of apoptotic HeLa cells in an OXPHOS-dependent culture condition.

**Figure supplement 3.** The initiation of caspase-3 activation occurs almost simultaneously with mitochondrial membrane potential loss.

cells, we compared the dynamics of intracellular ATP in PANX1-KO HeLa cells with those in wild-type cells during apoptotic progression under an OXPHOS-dominant culture condition. The cytosolic ATP level after the onset of caspase-3 activation dropped more quickly in the OXPHOS-dominant condition than in the glycolysis-dependent condition (*Figure 2—figure supplement 2*), probably

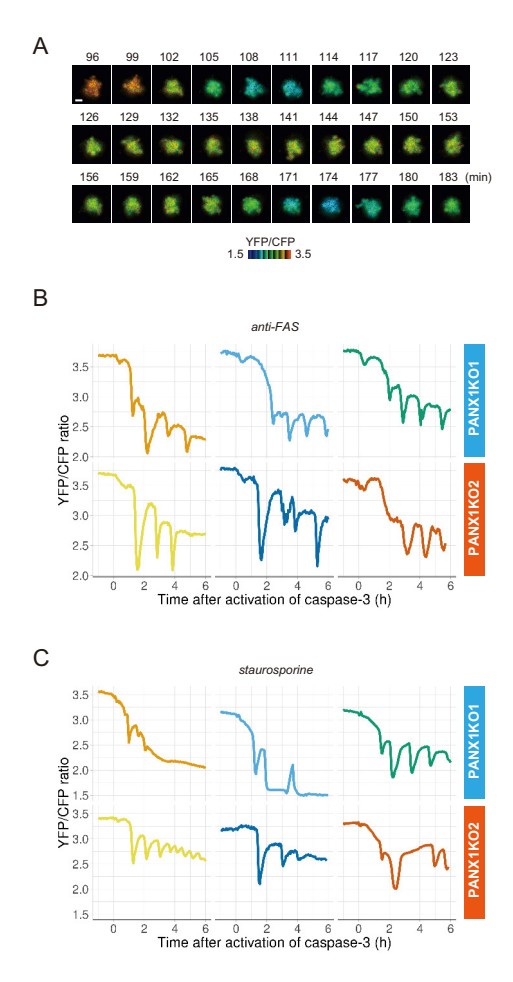

**Figure 3.** PANX1 knockout cells show fluctuations of cytosolic ATP levels after the activation of caspase-3. (**A**) Fluctuation of cytosolic ATP level of a PANX1-KO2 cell. Pseudocolored YFP/CFP ratio images of AT1.03CR are shown. Apoptosis was induced by anti-FAS and cycloheximide. Bar, 10 μm. (**B, C**) Representative traces of YFP/CFP ratios of AT1.03CR from single PANX1-KO cells from three biological replicates are shown. Apoptosis was induced by anti-FAS and cycloheximide (**B**), or staurosporine (**C**). Each trace was adjusted by setting the onset of caspase-3 activation as time = 0.

because mitochondrial membrane potential $\Delta\Psi_m$, which is required for mitochondrial ATP regeneration, is almost lost upon activation of caspase-3 (*Figure 2—figure supplement 3*). Even in this condition, knockout of PANX1 also suppressed the decrease in cytosolic ATP levels of the cells (*Figure 2—figure supplement 2*). This result indicates that PANX1 also promotes the intracellular ATP reduction of OXPHOS-dominant cells.

## Extracellular AMP suppresses the decrease in the cytosolic ATP

If efflux of adenine nucleotides through PANX1 channel causes the decrease in the cytosolic ATP concentrations of apoptotic cells, extracellular adenine nucleotides would suppress the decrease by counteracting the efflux of its cytosolic counterpart. We monitored cytosolic ATP dynamics in apoptotic cells in the presence of AMP, ADP, or ATP in the culture medium, and found that extracellular AMP suppressed the decrease in intracellular ATP levels of apoptotic cells, while extracellular ADP and ATP exhibited no or negligible effects (*Figure 5*). Thus, it is most likely that the intracellular ATP decrease of apoptotic cells is a result of a reduction in adenosine nucleotide pools inside apoptotic cells, which is caused, at least in part, by the release of AMP from the cells. This result is consistent with the previous reports that AMP constitutes a large part of adenine nucleotides released from apoptotic cells (*Yamaguchi et al., 2014*; *Boyd-Tressler et al., 2014*).

## PANX1 activity regulates free $Mg^{2+}$ dynamics, but not phosphatidylserine externalization, in apoptotic cells

Next, we investigated whether the PANX1 channel is involved in apoptotic events other than intracellular ATP reduction. First, we examined the $Mg^{2+}$ dynamics of apoptotic cells. $Mg^{2+}$ is an essential divalent cation in cells, required for various cellular processes, including the activity of endonucleases and the compaction of chromosomes during cell division (*Hartwig, 2001*; *Maeshima et al., 2018*). It is known that most of the intracellular ATP form a complex with $Mg^{2+}$, due to the high affinity of ATP for $Mg^{2+}$ (*Gupta and Moore, 1980*; *Grubbs, 2002*). Thus, ATP acts as a major intracellular chelator for $Mg^{2+}$. We hypothesized that the PANX1-dependent cytosolic ATP decrease might affect free $Mg^{2+}$ in apoptotic cells, and investigated the dynamics of free $Mg^{2+}$ in single apoptotic HeLa cells using a $Mg^{2+}$-sensing fluorescent probe MGH (*Matsui et al., 2017*). In wild-type cells, free $Mg^{2+}$ was transiently decreased after shrinkage of the cells. Subsequently, free $Mg^{2+}$ began to increase and often reached higher than the basal level. In contrast, the fluctuations in free $Mg^{2+}$ were significantly suppressed in apoptotic PANX1-KO cells (*Figure 6A*). Although the cause of the $Mg^{2+}$ decrease observed immediately after cell shrinkage is unclear, it might be possible that PANX1 transiently releases $Mg^{2+}$ from

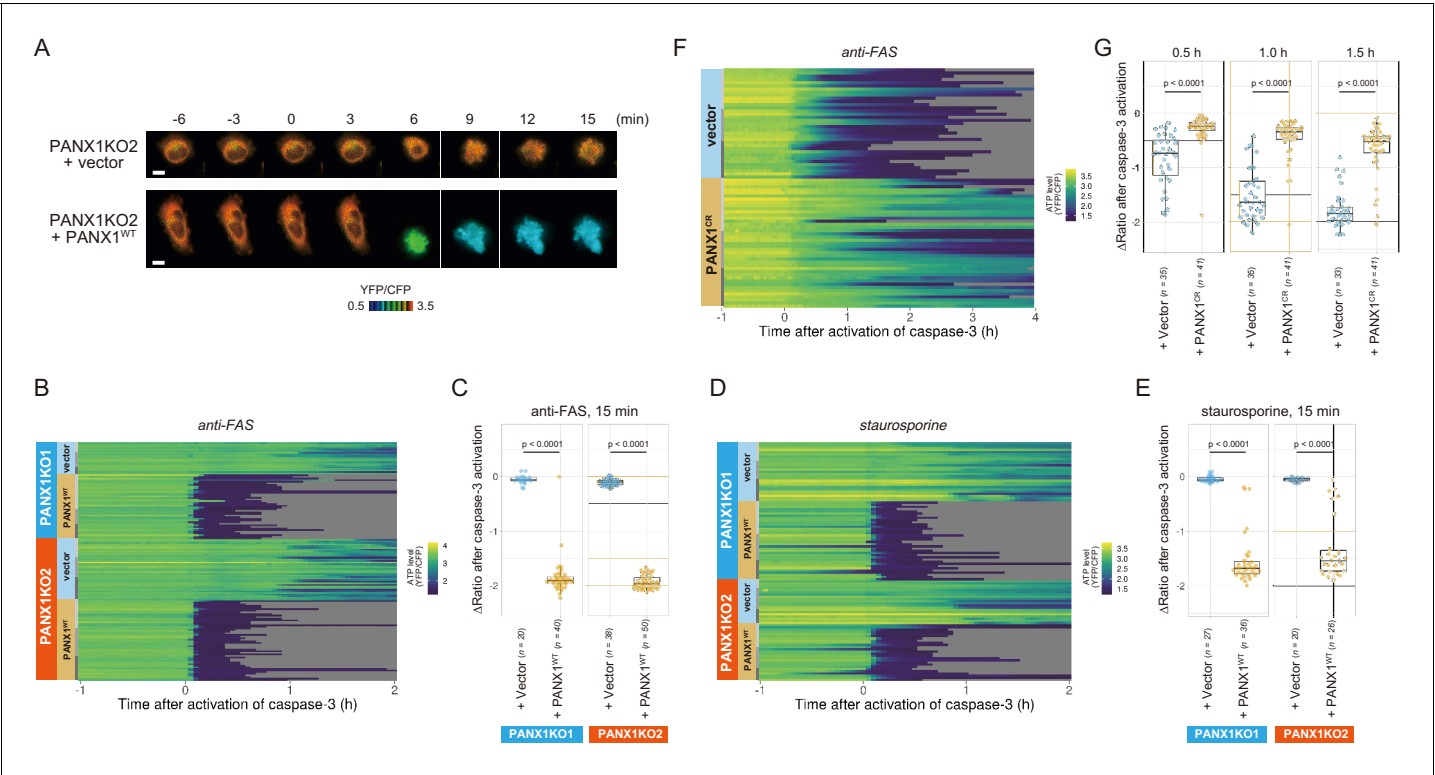

**Figure 4.** Exogenous expression of PANX1 alters cytosolic ATP dynamics of apoptotic cells. (**A**) Time-lapse images of the ATP level of PANX1-KO2 cells. Cells were transfected with either an empty vector (upper) or a vector expressing wild-type PANX1 (PANX1$^{WT}$) (lower). Pseudocolored ratio images of AT1.03CR are shown. (**B, D**) Single-cell ATP dynamics of PANX1-KO cells expressing exogenous PANX1$^{WT}$. Cells were transfected with either an empty vector or a vector expressing PANX1$^{WT}$. Apoptosis was induced by either anti-FAS and cycloheximide (B; 20 [PANX1-KO1 + vector], 40 [PANX1-KO1 + PANX1$^{WT}$], 38 [PANX1-KO2 + vector] and 50 [PANX1-KO2 + PANX1$^{WT}$] cells from three biological replicates), or staurosporine (D; 27 [PANX1-KO1 + vector], 36 [PANX1-KO1 + PANX1$^{WT}$], 20 [PANX1-KO2 + vector] and 26 [PANX1-KO2 + PANX1$^{WT}$] cells from three biological replicates). Each line represents the time course of the YFP/CFP ratio of AT1.03CR from a single cell, and was adjusted by setting the onset of caspase-3 activation as time = 0. (**C, E**) Effect of exogenous expression of PANX1$^{WT}$ on the decrease in cytosolic ATP levels. Changes in YFP/CFP ratios for 15 min after the onset of caspase-3 activation were calculated for each apoptotic cell. Apoptosis was induced by anti-FAS and cycloheximide (**C**), or staurosporine (**E**). Student's t-test. (**F**) Single-cell ATP dynamics of wild-type cells exogenously expressing a caspase-resistant mutant of PANX1. Wild-type HeLa cells were transfected with either an empty vector or a vector expressing a caspase-resistant mutant of PANX1 (PANX1$^{CR}$). Apoptosis was induced by anti-FAS and cycloheximide. Each line represents the time course of the YFP/CFP ratio of AT1.03CR from a single cell, and was adjusted by setting the onset of caspase-3 activation as time = 0 (35 [vector] and 41 [PANX1$^{CR}$] cells from three biological replicates). (**G**) Effect of exogenous expression of PANX1$^{CR}$ on the decrease in cytosolic ATP levels. Changes in YFP/CFP ratios at indicated time after the onset of caspase-3 activation were calculated for each apoptotic cell. Apoptosis was induced by anti-FAS and cycloheximide. P-values of Student's t-test are shown.

cytosol to extracellular space. The Mg$^{2+}$ increase in the second phase might be coupled with the decrease in ATP, an intracellular Mg$^{2+}$ chelator. Taken together, PANX1 regulates the dynamics of free Mg$^{2+}$ in apoptotic cells, likely in part by decreasing ATP concentrations inside cells. Second, we examined the role of PANX1 on the externalization of phosphatidylserine (PS) on plasma membrane, one of the hallmarks of apoptosis (*Elmore, 2007*; *Nagata, 2018*). Externalized PS functions as an 'eat-me' signal for phagocytosis of apoptotic cells by macrophages. We examined whether PANX1 channel affects the externalization of PS in the plasma membrane by quantifying the amount of externalized PS using fluorescently-labeled annexin-V. As a result, no significant difference in the externalized PS was observed between wild-type and PANX1-KO HeLa cells (*Figure 6B*), suggesting that neither the PANX1-dependent intracellular ATP reduction or PANX1 itself does not contribute to the externalization of PS during apoptotic progression.

## PANX1 activation prevents glucose expenditure by apoptotic cells

We showed above that dying apoptotic cells retained intracellular ATP levels for longer periods when the PANX1 channel is lost or suppressed. These observations suggest that an intracellular

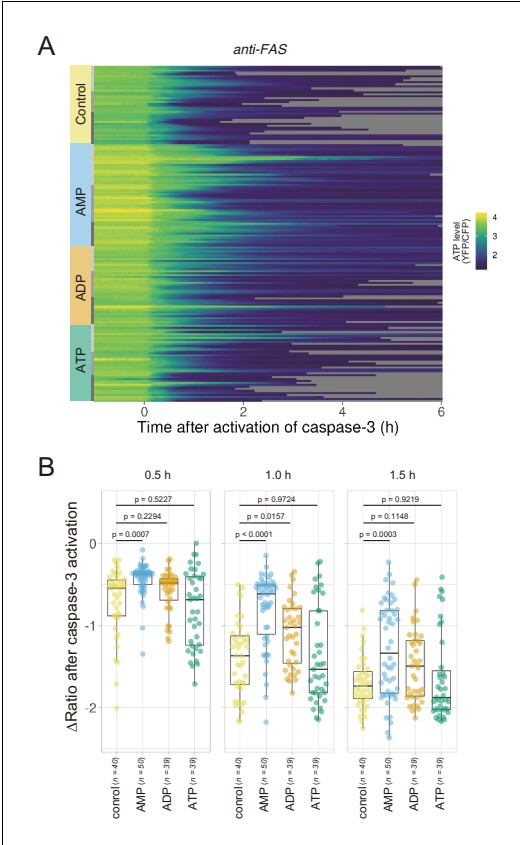

**Figure 5.** Extracellular AMP counteracts the decrease in the cytosolic ATP level of apoptotic cells. Wild-type HeLa cells expressing AT1.03CR and O-DEVD-FR were imaged in the presence of an adenine nucleotide (1 mM) in the culture medium. Apoptosis was induced by anti-FAS and cycloheximide. (**A**) Each line represents the YFP/CFP ratio of AT1.03CR from a single apoptotic cell, and was adjusted by setting the onset of caspase-3 activation as time = 0 (40 [control], 50 [AMP], 39 [ADP] and 39 [ATP] cells from three biological replicates). (**B**) Effect of the addition of extracellular nucleotide on the decrease in cytosolic ATP levels. Changes in FRET/CFP ratios at 0.5, 1.0, and 1.5 hr after the caspase-3 activation were calculated for each apoptotic cell. ANOVA followed by post-hoc Dunnett's test (versus control).

system for regenerating ATP from ADP and phosphate may still be active in dying apoptotic cells. In living cells, glycolysis and OXPHOS play prominent roles in the regeneration of ATP. To examine whether these ATP regenerating pathways are active, we treated PANX1-KO cells with either an inhibitor for glycolysis or OXPHOS after activation of caspase-3 while monitoring the dynamics of cytosolic ATP levels (*Figure 7A–D*). We used 2-deoxyglucose (2DG) or sodium oxamate, which are an inhibitor for hexokinase and lactate dehydrogenase, respectively, to inhibit glycolysis, whereas used oligomycin A, which is an inhibitor for $F_oF_1$-ATP synthase, to inhibit OXPHOS. Treatment with either 2DG or sodium oxamate induced a rapid decrease in intracellular ATP concentration (*Figure 7B,C*). In contrast, treatment with oligomycin A, an inhibitor of OXPHOS, seemed to have only a small effect on intracellular ATP dynamics under this condition (*Figure 7D*). These observations indicate that apoptotic processes do not disrupt the glycolytic system of the cells and that even dying apoptotic cells retain the ability to regenerate ATP by glycolysis. The rapid decrease in cytosolic ATP concentration of apoptotic PANX1-KO cells by inhibition of glycolysis also implies that at least some of the intracellular ATP-utilizing systems are active during apoptosis if sufficient intracellular ATP is present. It has been previously reported that apoptotic cells treated with a PANX1 inhibitor showed continuous and extensive blebbing (*Poon et al., 2014*), which is dependent on myosin ATPase (*Coleman et al., 2001*). Consistently, we also observed that PANX1-KO cells showed more extensive blebbing than wild-type cells during apoptosis (*Videos 1* and *2*). Moreover, forced ATP depletion of apoptotic PANX1-KO cells by 2DG leaded to the reduction of the blebbing of the cells (*Videos 3* and *4*). It is also likely that the reduction of ATP during apoptosis leads to the decrease in the activities of other ATPases because ATPase activity depends on the concentration of ATP. We expected that if the intracellular ATP concentration of apoptotic cells is not depleted, the cycle of ATP consumption and regeneration will continue, resulting in a continuous glucose consumption by the cells. To examine glycolytic activity in apoptotic cells, we quantified the consumption of glucose and the release of lactate by wild-type and PANX1-KO HeLa cells after induction of apoptosis. Both cells exhibited similar glucose consumption and lactate production rates when apoptosis was not induced (*Figure 7E*). Glucose consumption and lactate production by wild-type cells had almost ceased by 16 hr after induction of apoptosis, while those by PANX1-KO cells continued for at least 32 hr when apoptosis was induced by anti-FAS (*Figure 7F*). Trends of glucose consumption and lactate production by those cells were quite similar when apoptosis was induced by ultraviolet (*Figure 7G*). Thus, apoptotic cells with deficient PANX1 activity have a prolonged glycolytic activity compared to normal apoptotic cells. Taken together, activation of PANX1

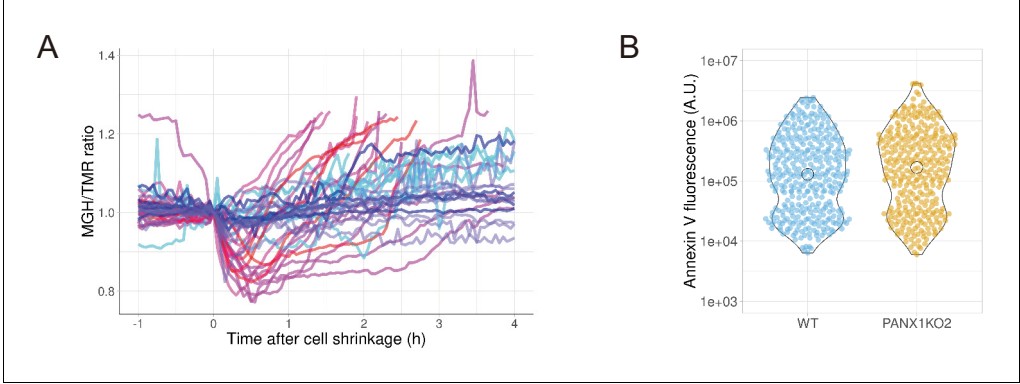

**Figure 6.** PANX1 regulates the cytosolic free Mg$^{2+}$ level, but not the externalization of phosphatidylserine, in apoptotic cells. (**A**) HeLa cells expressing Halo-tag were loaded with Halo-TMR and a magnesium indicator MGH (AM), followed by induction of apoptosis. Each trace represents a time course of normalized MGH/TMR fluorescence ratios of individual cell from three biological replicates. Timing of the initiation of cell shrinkage was defined as time = 0. Reddish and bluish traces represent wild-type cells (n = 22) and PANX1-KO2 cells (n = 20), respectively, from three biological replicates. Traces from different biological replicates were shown in different color codes. (**B**) Apoptosis of HeLa cells was induced by anti-FAS and cycloheximide in the presence of Alexa647-annexinV and propidium iodide. Fluorescence intensity of Alexa647-annexinV of each cell that does not show propidium iodide fluorescence was quantified using a fluorescent microscopy at 6 hr after induction of apoptosis (359 [WT] and 322 [PANX1-KO2] cells from three biological replicates).

The online version of this article includes the following source data for figure 6:

**Source data 1.** Numerical data used for panel B.

---

channels thwarts glucose expenditure of apoptotic cells, most likely by rapidly depleting intracellular ATP reserves (*Figure 8*).

## Discussion

In this study, we developed two genetically encoded FRET-based biosensors AT1.03CR and O-DEVD-FR, which enabled dual imaging of both ATP levels and caspase-3 activities during the apoptotic process at the single-cell level. This method allowed us to analyze the single-cell dynamics of cytosolic ATP level after caspase-3 activation, which occurs at different times in different cells. It was clearly shown that the cytosolic ATP level remained almost constant until caspase-3 was activated (*Figure 1C,D*). We did not observe any profound immediate ATP changes upon induction of apoptosis (*Figure 1—figure supplement 2*), in contrast to the previous intracellular ATP analysis from populations of cells using firefly luciferase that has suggested acute ATP production when apoptosis is induced (*Zamaraeva et al., 2005*). This discrepancy is not clear at present. Once caspase-3 was activated, cytosolic ATP level of the cells started to drop and was typically depleted within 2 hr (*Figure 1C,D*). Previous studies have shown that sufficient levels of intracellular ATP are required for progression of apoptosis (*Eguchi et al., 1997*; *Hu et al., 1999*; *Leist et al., 1997*; *Li et al., 1997*; *Zamaraeva et al., 2005*) and that pre-reduction of intracellular ATP inhibits the activation of caspase-3 (*Zamaraeva et al., 2005*). Moreover, apoptosome formation has been shown to require dATP/ATP in vitro (*Hu et al., 1999*; *Li et al., 1997*). Taken together, it is likely that maintenance of high intracellular ATP is critical to activate caspases. Further analysis on cytosolic ATP dynamics demonstrated that PANX1 channels play a major role in intracellular ATP depletion in apoptotic cells (*Figures 2* and *4*, S4 and S5). It has been reported that PANX1 releases adenine and uridine nucleotides upon activation by cleavage of the C-terminal cytosolic region of the protein by effector caspases (caspase-3 and 7) and that the released nucleotides act as 'find-me' signals for attracting macrophages, which engulf apoptotic cells (*Chekeni et al., 2010*; *Yamaguchi et al., 2014*). In this study, we show that the caspase-dependent cleavage of PANX1 is also crucial for the intracellular ATP depletion, meaning that intracellular ATP depletion in apoptotic cells is a 'programmed' process rather than a passive phenomenon. The depletion of intracellular ATP is most likely the result of the decrease in intracellular adenine nucleotide pool, caused by the release of AMP from the cytosol

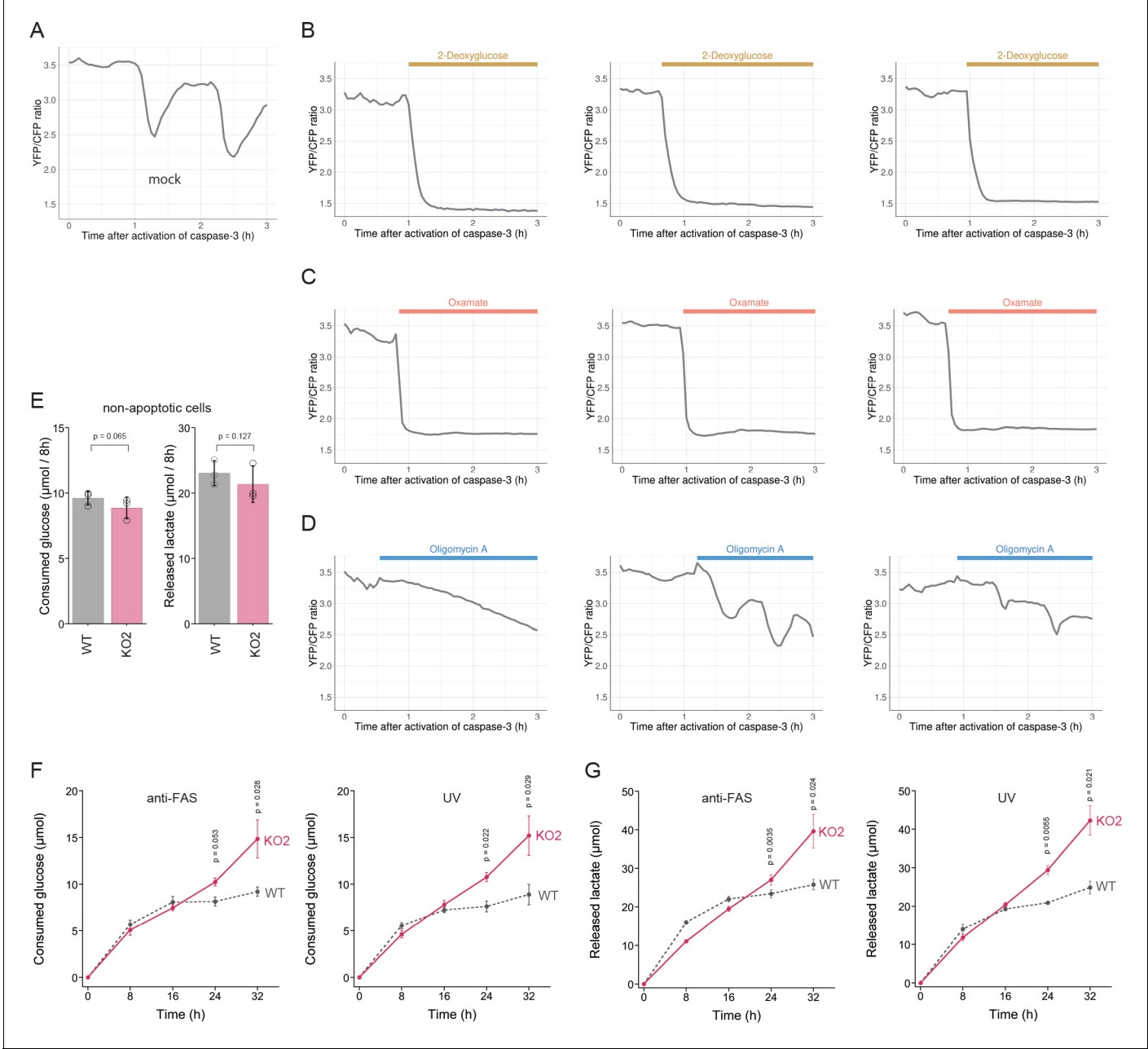

**Figure 7.** PANX1 is required for suppressing consumption of glucose by apoptotic cells. (A–D) Effect of metabolic inhibitors on cytosolic ATP of apoptotic cells. Apoptosis of PANX1-KO2 cells expressing AT1.03CR and O-DEVD-FR were induced by anti-FAS and cycloheximide. After activation of caspase-3, the cells were untreated (A) or treated either with 10 mM 2-deoxyglucose (B), 50 mM sodium oxamate (C), or 1 μg/ml oligomycin A. Each trace represents the YFP/CFP ratio of AT1.03CR from a single apoptotic cell, and was adjusted by setting the onset of caspase-3 activation as time = 0. A period of inhibitor treatment was indicated as a bar on each plot. Representative traces from three biological replicates are shown. (E) Glycolytic activity of living wild-type and PANX1-KO cells. Consumption of glucose (left) and release of lactate (right) in 8 hr is shown. (F–G) Glycolytic activity of apoptotic cells. Apoptosis was induced either by anti-FAS/cycloheximide (left) or ultraviolet (right) at time = 0. Consumed glucose (F) and released lactate (G) by wild-type Hela cells (black, dashed line) and PANX1-KO2 cells (red, solid line) are shown. Means ± s.d. (three biological replicates) and p-values of paired t-test were shown (E–G).

The online version of this article includes the following source data for figure 7:

**Source data 1.** Numerical data used for panel E.
**Source data 2.** Numerical data used for panel F.
**Source data 3.** Numerical data used for panel G.

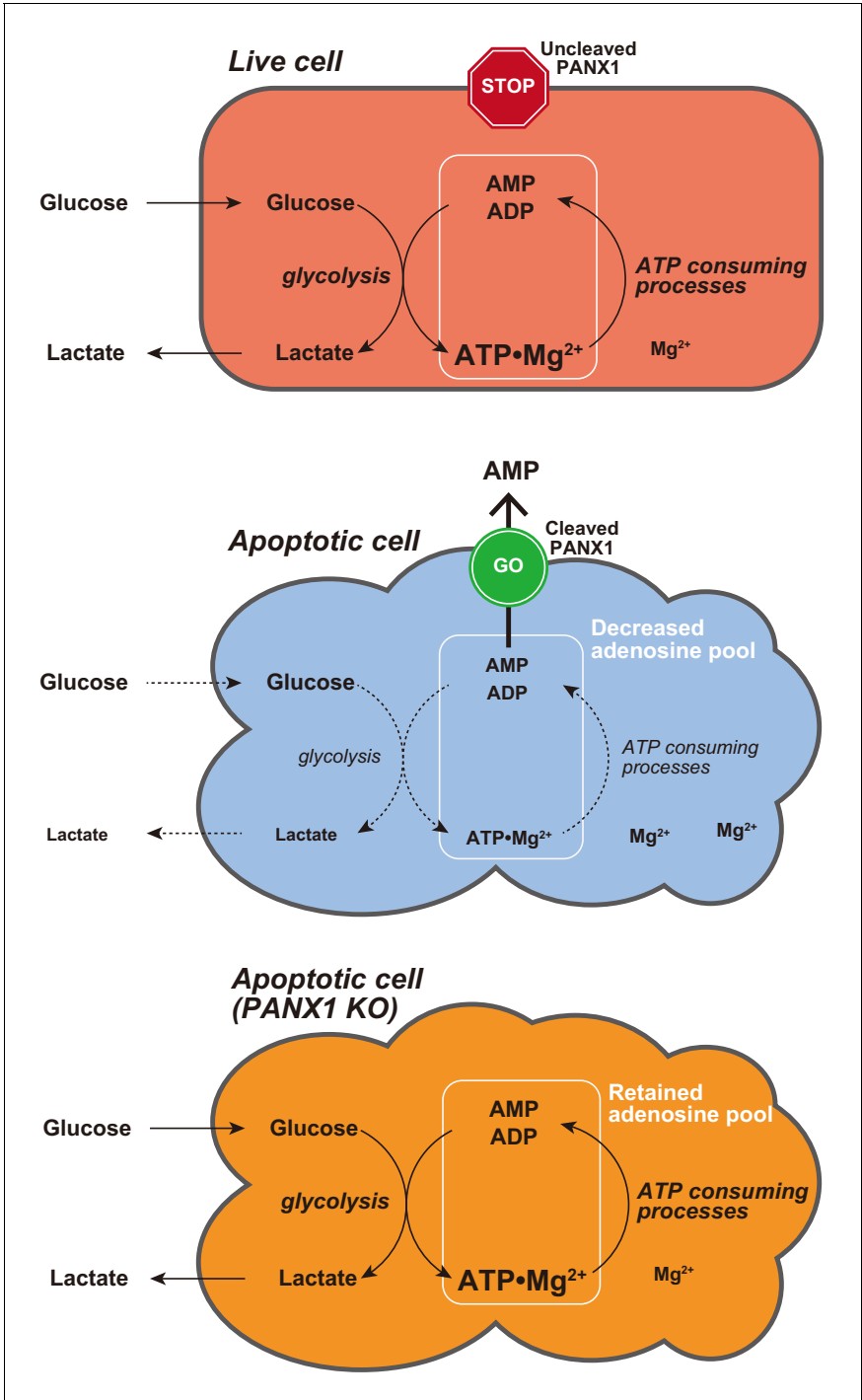

**Figure 8.** Proposed model for PANX1-dependent control of the cytosolic ATP level and glucose consumption of apoptotic cells.

to the extracellular space (*Figure 8*). It should be noted that the intracellular ATP decrease in apoptotic cells could not be completely stopped by knockout of PANX1 (*Figure 2*). Therefore, there must be an additional mechanism of decreasing the intracellular ATP level in apoptotic cells, which is also likely to contribute to the kinetic variations in cytosolic ATP reduction among individual apoptotic cells (*Figure 2C–F*, *Figure 2—figure supplement 1*).

Metabolism is one of the major biological activity of living systems. The metabolic activity of cells is an indicator of 'the state of being alive'. In other words, in order for a cell to 'die a complete

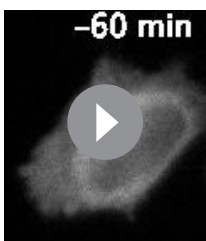

**Video 1.** Time-lapse fluorescent movie of an apoptotic wild-type HeLa cell. YFP (FRET) fluorescence from ATeam is shown. Apoptosis was induced by anti-FAS and cycloheximide. Time = 0 indicates the onset of caspase-3 activation.

https://elifesciences.org/articles/61960#video1

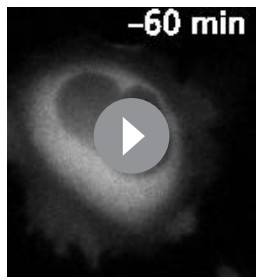

**Video 2.** Time-lapse fluorescent movie of an apoptotic PANX1-KO HeLa cell. The same experimental conditions as *Figure 1—figure supplement 1*.

https://elifesciences.org/articles/61960#video2

death', metabolism must stop. However, it has not been well understood how metabolism of dead cells ceases. Our results showed that PANX1 plays a major role in stopping glycolysis, a central part of metabolism, of apoptotic cells, most likely through facilitating ATP loss. The suppression of glucose expenditure by apoptotic cells may benefit surrounding live cells because the resource of glucose for surrounding live cells would be limited if dying cells continued to consume glucose. Besides, macrophages that eat apoptotic cells may encounter a risk of energy deprivation if the eaten cells actively consume glucose inside the macrophages.

Because the removal of apoptotic cells by macrophages is critical to suppress inflammation and autoimmune diseases (*Nagata, 2018*), stimulating local macrophages by ATP and related nucleotides released through PANX1 channels should be a biologically important process in apoptosis (*Chekeni et al., 2010*). However, it is not clear why apoptotic cells use ATP and related nucleotides as major chemoattractants for macrophages over others, such as lysophosphatidylcholine (*Ousman and David, 2000*). Although it is difficult to say whether the primary function of PANX1 opening in apoptosis is to deplete intracellular ATP reserves or to release chemoattractant, or both, one possible scenario may be that cells first developed the function of PANX1 for depleting intracellular ATP, followed by effective utilization of the released adenine nucleotides as chemoattractants.

How does the intracellular ATP concentration change in other types of cell death, such as necrosis and pyroptosis? Because a forced decrease of intracellular ATP levels reportedly switches the cell death fate from apoptosis to necrosis, it is believed that the intracellular ATP level will decrease in the early stage of necrosis (*Eguchi et al., 1997*; *Leist et al., 1997*; *Tsujimoto, 1997*). However, a precise investigation of when and how the intracellular ATP level changes in the necrotic process is still lacking, especially at single-cell resolution. In

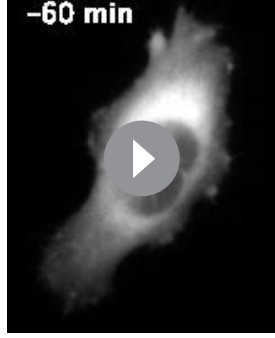

**Video 3.** Time-lapse florescent movie of an untreated apoptotic PANX1-KO HeLa cell. Apoptosis was induced by anti-FAS and cycloheximide. Time = 0 indicates the onset of caspase-3 activation.

https://elifesciences.org/articles/61960#video3

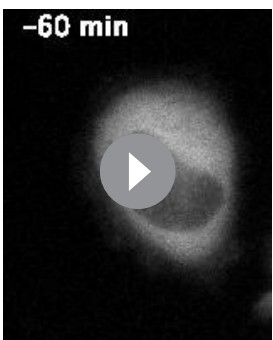

**Video 4.** Time-lapse fluorescent movie of a 2DG-treated apoptotic PANX1-KO HeLa cell. Apoptosis was induced by anti-FAS and cycloheximide. Time = 0 indicates the onset of caspase-3 activation. 2DG was added between 60 and 63 min.

https://elifesciences.org/articles/61960#video4

pyroptosis, gasdermin-D has been reported to form pores as large as 10 nm in diameter in the plasma membrane and to release adenine nucleotides (*Liu et al., 2016*; *Russo et al., 2016*). However, it is not clear whether a drop in the intracellular ATP level coincides with the gasdamin-D pore formation. The techniques used in this study might be useful for uncovering the mechanism of how ATP decreases in other types of cell death.

# Materials and methods

## Key resources table

| Reagent type (species) or resource | Designation | Source or reference | Identifiers | Additional information |
|---|---|---|---|---|
| Cell line (*H.sapiens*) | HeLa | Shin Yonehara Lab | | Authenticated by STR profiling; mycoplasma tested |
| Cell line (*H.sapiens*) | HeLa-PANX1-KO1 | This study | | PANX1-knockout line |
| Cell line (*H.sapiens*) | HeLa-PANX1-KO2 | This study | | PANX1-knockout line |
| Cell line (*H.sapiens*) | SW480 | ATCC | ATCC: CCL-228; RRID:CVCL_0546 | |
| Cell line (*H.sapiens*) | SW480-PANX1-KO | This study | | PANX1-knockout line |
| Antibody | anti-FAS (Mouse monoclonal) | Molecular biology laboratory | Cat. #:SY-001, RRID:AB_591016 | |
| Antibody | anti-His tag (Mouse monoclonal) | R and D Systems | Cat. #: MAB050-100 | |
| Antibody | anti-PANX1 (Rabbit monoclonal) | Cell Signalling | Cat# 91137, RRID:AB_2800167 | WB (1:1000) |
| Antibody | HRP-conjugated anti-β-actin (Mouse monoclonal) | Santa Cruz Biotechnology | Cat# sc-47778 HRP, RRID:AB_2714189 | WB (1:1000) |
| Recombinant DNA reagent | pcDNA3.1-AT1.03 (plasmid) | *Imamura et al., 2009* | | Original ATP biosensor |
| Recombinant DNA reagent | pcDNA3.1-AT1.03CR (plasmid) | This paper | | Caspase-resistant ATP biosensor |
| Recombinant DNA reagent | pcDNA3.1-O-DEVD-FR (plasmid) | This paper | | Caspase-3 biosensor |
| Recombinant DNA reagent | IRAK027A14 (plasmid) | Riken Bioresource Center | Cat. #: HGX010814 | Human pannexin-1 cDNA clone |
| Recombinant DNA reagent | pIRES2-Sirius (plasmid) | This paper | | |
| Recombinant DNA reagent | pIRES2-Sirius-PANX1 (plasmid) | This paper | | Wild-type human PANX1 |
| Recombinant DNA reagent | pIRES2-Sirius-PANX1CR (plasmid) | This paper | | Caspase-resistant human PANX1 mutant |
| Recombinant DNA reagent | pcDNA4-SCAT3.1 (plasmid) | *Nagai and Miyawaki, 2004* | | Caspase-3 biosensor |

*Continued on next page*

*Continued*

| Reagent type (species) or resource | Designation | Source or reference | Identifiers | Additional information |
|---|---|---|---|---|
| Recombinant DNA reagent | Pannexin-1 CRISPR/Cas9 KO Plasmid (h) | Santa Cruz Biotechnology | Cat. #: sc-401635 | |
| Peptide, recombinant protein | TRAIL, human | R and D Systems | Cat. #: 375-TL-010 | |
| Commercial assay or kit | Glucose CII-test Wako | Wako Pure Chemicals | Cat. #: 300167002 | |
| Commercial assay or kit | Lactate Assay Kit-WST | Dojindo | Cat. #: L256 | |
| Chemical compound, drug | MGH | *Matsui et al., 2017* | | |
| Chemical compound, drug | Halo-TMR | Promega | Cat. #: G8251 | |
| Chemical compound, drug | Annexin-V, Alexa647 conjugate | Molecular Probes | Cat. #: A23204 | |
| Chemical compound, drug | Tetramethylrhodamine ethyl ester | Molecular Probes | Cat. #: T669 | |
| Software, algorithm | Metamorph | Molecular Devices | RRID:SCR_002368 | |

## Materials

Oligomycin A, ATP, ADP, and AMP were obtained from Sigma-Aldrich. Solutions of ATP, ADP, and AMP were neutralized with sodium hydroxide before use. Anti-FAS antibody was from Molecular Biology Laboratory (Nagoya, Japan). Cycloheximide (CHX) was from Calboichem. Tetramethylrhodamine ethyl ester (TMRE) and alexa647-annexin-V were purchased from Molecular Probes. Recombinant human Trail and anti-His tag antibody were purchased from R and D. Trail was dissolved in DPBS(-) supplemented with 0.1% BSA at a concentration of 20 µg/mL, and then stored at −80 ˚C. Anti-His antibody was dissolved in DPBS(-) at a concentration of 1 mg/mL, and then stored at −80 ˚C. 2-Deoxyglucose were from Wako Pure Chemicals (Osaka, Japan). Other chemicals were purchased from Nacalai Tesque (Kyoto, Japan) unless otherwise noted.

## Mammalian cell culture and gene knock out

The HeLa cell line was a kind gift from Prof. Shin Yonehara. The cell line was authenticated by short-tandem repeat profiling and was checked for mycoplasma infection. The cells were grown in Dulbecco's modified Eagle's medium (DMEM, 1 g/L glucose; Nacalai Tesque) supplemented with 10% fetal bovine serum (FBS; Sigma-Aldrich). The SW480 cell line was obtained from American Type Culture Collection. The cells were cultured in Leibovitz's L-15 medium (Wako Pure Chemicals, Japan) supplemented with 10% FBS. Apoptosis of HeLa cells was initiated either by adding anti-FAS antibody (125 ng/mL) and CHX (10 µM) or by adding staurosporine (1 µM). Apoptosis of SW480 cells was initiated by adding Trail (50 ng/mL) and anti-His tag antibody (1 µg/mL). Knockout of PANX1 gene was carried out using pre-designed PANX1-KO CRISPR-Cas9 plasmids (Santa Cruz Biotechnology). Briefly, cells were transfected with CRISPR-Cas9 plasmids using PEI-Max (Polysciences Inc) as described previously (*Morciano et al., 2020*). After 2 days, each individual cell with strong GFP fluorescence was sorted into a well in 96-well plates using a cell sorter (SH800S, Sony), and then was cultured. Knock-out of PANX1 gene in each cultured line was verified by western blotting and by sequencing of the targeted region of the genomic DNA.

## Plasmids

The expression vector for the caspase-resistant AT1.03 (pcDNA-AT1.03CR) was constructed by introducing caspase-resistant mutations (D242N/D339G) into pcDNA-AT1.03 (*Imamura et al., 2009*) using PCR-based mutagenesis. O-DEVD-FR cDNA was constructed by fusing mKOκ and mKate2 (V94S) through a Gly-Gly-Asp-Glu-Val-Asp-Gly-Thr linker using PCR. The amplified cDNA was cloned between XhoI and HindIII sites of pcDNA3.1(-) (Thermo Scientific) to obtain a mammalian expression vector pcDNA-O-DEVD-FR. Human PANX1 cDNA (Riken Bioresource Center) was amplified by PCR, and was cloned between XhoI and EcoRI sites of pIRES2-Sirius, a custom made vector, in which EGFP cDNA of pIRES2-EGFP (Clontech) was replaced by Sirius fluorescent protein cDNA (*Tomosugi et al., 2009*), to obtain a pIRES2-Sirius-hPANX1 plasmid. Caspase-resistant mutations (D376A/D379A) in PANX1 were introduced by PCR-based mutagenesis.

## Fluorescence imaging of ATP levels and caspase-3 activities

Cells were transfected with the AT1.03CR plasmid and the O-DEVD-FR plasmid using PEI-Max as described previously (*Morciano et al., 2020*). For exogenous expression of PANX1, the plasmid encoding PANX1 cDNA was co-transfected with the AT1.03CR and the O-DEVD-FR plasmids. One day after transfection, cells were trypsinized and plated on a collagen-coated glass-bottom 4-compartment dish (0.16–0.19 mm thick; Greiner). Two days after transfection, cells cultured in phenol red-free DMEM supplemented with 10% FBS (HeLa) or phenol red-free Leibovitz's L-15 medium supplemented with 10% FBS (SW480) were subjected to imaging. For OXPHOS-dependent cell culture, phenol red- and glucose-free DMEM (Gibco) supplemented with 10% FBS, 10 mM sodium lactate (Sigma-Aldrich), and 10 mM sodium dichloroacetate (an inhibitor of pyruvate dehydrogenase kinase, Sigma-Aldrich) was used. Cells were visualized with a Ti-E inverted microscope (Nikon, Tokyo, Japan) using a Plan Apo 40×, 0.95 numerical aperture, dry objective lens (Nikon). Cells were maintained on a microscope at 37°C with a continuous supply of a 95% air and 5% carbon dioxide mixture by using a stage-top incubator (Tokai Hit). All filters used for fluorescence imaging were purchased from Semrock (Rochester, NY): for dual-emission ratio imaging of AT1.03CR, an FF01-438/24 excitation filter, an FF458-Di02 dichroic mirror, and two emission filters (an FF02-483/32 for CFP and an FF01-542/27 for YFP); dual-emission ratio imaging of O-DEVD-FR, an FF01-543/22 excitation filter, an FF562-Di02 dichroic mirror, and two emission filters (an FF01-585/22 for mKOκ and an FF01-660/52 for mKate2). Cells were illuminated using a 75 W xenon lamp through 25% and 12.5% neutral density filters. Fluorescence emissions from cells were imaged using a Zyla4.2 scientific CMOS camera (Andor Technologies). The microscope system was controlled by NIS-Elements software (Nikon). Image analysis was performed using MetaMorph software (Molecular Devices). First, a background fluorescence intensity, which was measured from a region within image where no cell exist, was subtracted from an entire image. Next, the intensity of a donor fluorophore (YFP or mKate2) of a cell was divided by the intensity of an acceptor fluorophore (CFP or mKOκ) to obtain the emission ratio. The detailed method for the image analysis has been described previously (*Morciano et al., 2020*). Because fluorescent biosensors were abruptly dissipated from the cell when the plasma membrane of the apoptotic cell was collapsed, data after fluorescence values from cells were less than half of the maximum were excluded from the analysis. Single-cell ATP traces were visualized using PlotTwist (*Goedhart, 2020*). Plots of single-cell ATP changes were generated using PlotsOfData (*Postma and Goedhart, 2019*).

## Fluorescence imaging of caspase-3 activity and $\Delta\Psi_m$

HeLa cells were transfected with the pcDNA-SCAT3.1 plasmid (*Nagai and Miyawaki, 2004*) using Lipofectamine 2000 (Thermo Scientific). One day after transfection, cells were trypsinized and plated on a collagen-coated glass-bottom dish (0.16–0.19 mm thick; MatTek). Two days after transfection, the medium was replaced by phenol red-free DMEM containing 10% FBS and 50 nM TMRE. Then, the cells were visualized with a Ti-E inverted microscope (Nikon, Tokyo, Japan) using a Plan Apo 40×, 0.95 numerical aperture, dry objective lens (Nikon). Cells were maintained on a microscope at 37°C with a continuous supply of a 95% air and 5% carbon dioxide mixture by using a stage-top incubator (Tokai Hit). All filters used for fluorescence imaging were purchased from Semrock (Rochester, NY): for dual-emission ratio imaging of SCAT3.1 biosensors, an FF01-438/24 excitation filter, an FF458-Di02 dichroic mirror, and two emission filters (an FF02-483/32 for CFP and an FF01-542/27

for YFP); for imaging of TMRE, an FF01-562/40 excitation filter, an FF593-Di02 dichroic mirror, and an FF01-641/75 emission filter. Cells were illuminated using a 75 W xenon lamp through 25 and 12.5% neutral density filters. Fluorescence emissions from cells were imaged using a Zyla4.2 scientific CMOS camera (Andor Technologies). The microscope system was controlled by NIS-Elements software (Nikon). Image analysis was performed using MetaMorph software (Molecular Devices).

## Fluorescence imaging of free $Mg^{2+}$

$Mg^{2+}$ imaging was performed using a synthetic $Mg^{2+}$ indicator MGH (*Matsui et al., 2017*). HeLa cells maintained in 10% FBS in DMEM (Invitrogen) at 37°C under 5% $CO_2$ were transfected with the plasmid pcDNA-3.1-(+)-Halo-Tag using PEI-Max (Polysciences). After 48 hr, the cells were washed twice with HBSS and incubated with 5 µM MGH(AM) for 30 min, then 50 nM Halo-TMR for 100 min. After washing the cells twice with HBSS, the medium was replaced by phenol red-free DMEM containing 10% FBS. After 4 hr, the cells were washed with HBSS. Imaging of the cells was started just after replacing the medium by phenol red-free DMEM containing 10% FBS, 50 ng/mL anti-FAS and 10 µM cycloheximide. Cells were visualized with a Ti-E inverted microscope using a Plan Apo 40×, 0.95 numerical aperture, dry objective lens (Nikon). Cells were maintained on a microscope at 37°C with a continuous supply of a 95% air and 5% carbon dioxide mixture by using a stage-top incubator (Tokai Hit). For imaging of MGH, an FF01-497/16 excitation filter, an FF516-Di01 dichroic mirror and an FF01-535/22 emission filter were used. For imaging of TMR, an FF01-562/40 excitation filter, an FF593-Di02 dichroic mirror, and an FF01-641/75 emission filter were used. All filters were purchased from Semrock.

## Fluorescence imaging of phosphatidylserine

HeLa cells were cultured on a 35 mm glass-bottom culture dish. At 6 hr before imaging, the medium was replaced by 2 mL of phenol red-free DMEM supplemented with FBS (10%), $CaCl_2$ (1 mM), propidium iodide (1 µg/mL), Annexin-V-Alexa647 (10 µL), anti-FAS (250 ng/mL) and CHX (10 µM). Cells were visualized with a Ti-E inverted microscope (Nikon, Tokyo, Japan) using a Plan Apo 20×, 0.75 numerical aperture, dry objective lens (Nikon). All filters used for fluorescence imaging were purchased from Semrock (Rochester, NY): for imaging of propidium iodide, an FF01-504/12 excitation filter, an FF593-Di02 dichroic mirror, and an FF01-562/40 emission filter; for imaging of Annexin-V-Alexa647, an FF02-628/40 excitation filter, an FF660-Di02 dichroic mirror, and an FF01-692/40 emission filter. Fluorescence emissions from cells were imaged using a Zyla4.2 scientific CMOS camera (Andor Technologies). Image analysis was performed using MetaMorph software (Molecular Devices). Fluorescence intensity of Alexa647 within a whole-cell area of each cell that was shrunk but did not exhibit propidium iodide fluorescence was quantified.

## Glucose and lactate assay

HeLa cells ($1.5 \times 10^6$) were cultured in 60 mm dish in DMEM (1 g/L glucose) supplemented with 10% FBS. After 24 hr, the medium was replaced by HBSS. Subsequently, apoptosis was induced either by replacing the medium with 4 mL of phenol red-free DMEM (1 g/L glucose) supplemented with 10% FBS, 250 ng/mL anti-FAS antibody and 10 µM cycloheximide, or by irradiating the cells with 20 mJ UV-C, followed by replacing the medium with 4 mL of phenol red-free DMEM (1 g/L glucose) supplemented with 10% FBS. Small aliquots of culture medium from the cell cultures were sampled at defined intervals. After centrifugation at 3000 x g for 3 min at 4 °C, the supernatant from each sample was stored at −30°C until the glucose and lactate quantification assay. Glucose and lactate concentrations of the aliquots were determined using Glucose CII-test Wako (Wako Pure Chemicals, Osaka, Japan) and Lactate Assay Kit-WST (Dojindo, Kumamoto, Japan), respectively.

## Western blotting

Cleavage of ATeam in apoptotic cells was examined by western blotting. Briefly, apoptosis was induced in HeLa cells, which were previously transfected with pcDNA-AT1.03 or pcDNA-AT1.03CR, by adding anti-FAS antibody and CHX. The cells were harvested 12 hr after anti-FAS stimulation and lysed. After separation of the cell lysate by SDS-PAGE, ATeam and actin proteins were detected by western blotting using a polyclonal anti-GFP antibody (Invitrogen), which cross-reacts with both CFP and YFP, and a monoclonal anti-actin antibody (Millipore), respectively. Protein expression levels of

PANX1 were examined by western blotting using a rabbit monoclonal antibody against human PANX1 (D9M1C, Cell Signaling). Actin was also detected as a control by western blotting using a mouse monoclonal anti-β-actin antibody (Santa Cruz). Horseradish peroxidase (HRP)-labeled anti-mouse or anti-rabbit IgG antibody (GE healthcare) was used as a secondary antibody. Chemi-Lumi One L reagent (Nacalai tesque) was used as a HRP substrate. A LAS4000 imager (GE healthcare) was used to detect the luminescence.

## Acknowledgements

We thank Shin Yonehara for technical advice and for providing HeLa cells, Hiroyuki Noji, Takeharu Nagai and Kenta Saito for technical advice, and James Alan Hejna and Kunio Takeyasu for critical assessment of this manuscript. We are grateful to Takeharu Nagai and Atsushi Miyawaki for sharing the SCAT3.1 plasmid. This work was supported by JSPS KAKENHI Grant Number 22687011, 24657101 and 16K14709 (to HI), and by the Platform Project for Supporting Drug Discovery and Life Science Research (Platform for Dynamic Approaches to Living Systems) from the Ministry of Education, Culture, Sports, Science and Technology (MEXT) and the Japan Agency for Medical Research and Development (AMED).

## Additional information

### Funding

| Funder | Grant reference number | Author |
|---|---|---|
| Japan Society for the Promotion of Science | 22687011 | Hiromi Imamura |
| Japan Society for the Promotion of Science | 24657101 | Hiromi Imamura |
| Japan Society for the Promotion of Science | 16K14709 | Hiromi Imamura |

The funders had no role in study design, data collection and interpretation, or the decision to submit the work for publication.

### Author contributions

Hiromi Imamura, Conceptualization, Data curation, Formal analysis, Funding acquisition, Writing - original draft, Project administration; Shuichiro Sakamoto, Conceptualization, Data curation, Formal analysis, Writing - review and editing; Tomoki Yoshida, Data curation, Writing - review and editing; Yusuke Matsui, Data curation, Formal analysis, Writing - review and editing; Silvia Penuela, Dale W Laird, Kazuya Kikuchi, Resources, Writing - review and editing; Shin Mizukami, Resources, Data curation, Formal analysis, Writing - review and editing; Akira Kakizuka, Conceptualization, Writing - review and editing

### Author ORCIDs

Hiromi Imamura (iD) https://orcid.org/0000-0002-1896-0443
Silvia Penuela (iD) http://orcid.org/0000-0003-4829-5517
Akira Kakizuka (iD) http://orcid.org/0000-0003-3513-0334

### Decision letter and Author response

Decision letter https://doi.org/10.7554/eLife.61960.sa1
Author response https://doi.org/10.7554/eLife.61960.sa2

## Additional files

### Supplementary files

• Transparent reporting form

## Data availability

All data analyzed during this study are included in the manuscript. Source data files have been provided for Figure 6 and 7. DNA and amino acid sequences for a developed caspase-3 biosensor is presented in Figure 1-figure supplement 3.

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
