## [Decision Letter]

**Acceptance summary:**

By designing an assay that allows continuous real-time dual imaging of intracellular ATP level and caspase-3 activity in single apoptotic cultured human cells, the authors have gained detailed knowledge of the kinetics of the decline in cellular ATP relative to caspase activation. This allowed the authors to nail down that adenine nucleotide via caspase activation of pannexin-1 channels early in apoptosis not only recruits phagocytes to the apoptotic cells, but also makes a major contribution to the eventual arrest of energy-dependent functions in the dying cells.

**Decision letter after peer review:**

[Editors’ note: the authors submitted for reconsideration following the decision after peer review. What follows is the decision letter after the first round of review.]

Thank you for submitting your work entitled "Pannexin-1 mediates programmed intracellular ATP decrease during apoptosis" for consideration by *eLife*. Your article has been reviewed by four peer reviewers, one of whom is a member of our Board of Reviewing Editors, and the evaluation has been overseen by a Senior Editor. The following individual involved in review of your submission has agreed to reveal their identity: George Dubyak (Reviewer #3).

Our decision has been reached after consultation between the reviewers. Based on these discussions and the individual reviews below, we regret to inform you that your work will not be considered further for publication in *eLife*.

Your manuscript "Pannexin-1 mediates programmed intracellular ATP decrease during apoptosis" has been examined by four expert reviewers whose detailed comments are enclosed. As you can see, the reviewers were very appreciative of the means that you have established for monitoring the dynamics of changes in the cytosolic ATP concentration on a single-cell level during apoptosis. It is certainly a valuable addition to the analytical toolbox applied in the study of cell death. The reviewers were reserved, however, about the fact that too much of the paper (including its title) concerns the finding that Pannexin-1 mediates the decrease of ATP in apoptosis – a finding that has been already documented before. They also raised many questions and requests about the evidence for the new biological insights that the paper aimed to present.

Since the breath of the requests made by the reviewers is too wide to allow addressing them within the two month period allowed by *eLife*, I am unfortunately unable to recommend further handling of current submission. However, we do hope that you will submit the paper anew to *eLife* after addressing the comments of the reviewers.

A simpler and quicker alternative would be to resubmit the paper, not as Research Article, but in the Tools and Resources category (after adding more detailed description of the ATP monitoring technique). This category is appropriate for a new technique that will be generally useful to a field, and although new biological insight is not required in this category you should be able also to include your major observations in the "Tools and Resources" format.

Reviewer #1:

Overall this is an interesting study that adds incremental insight to the original story from Chekeni et al., 2010, follow up study by Sandilos et al., 2012, and other more recent work. The use of imaging to assay ATP levels creates a potential for revealing spatial information but this potentially novel aspect is unfortunately not fully realized. Most importantly, clear interpretations of some of the key pieces of imaging data are hampered by the lack of controls, including controls demonstrating that the loss of signal is not a loss of cell area/cell collapse. Further, rather than relying primarily on the use of the blocker probenecid for a large portion of the study, the systematic use of a more complete genetic deletion model like Crispr Cas9, which should be readily feasible in the Hela model, would also make the interpretation and model more convincing. There are also some kinks in the logic/model that could be smoothened out (clarified) by more detailed explanations. In summary, there is potential here, but the manuscript seems premature.

Detailed comments:

Abstract: I'm not sure this sentence in the Abstract makes sense (nor is it substantiated by the data) – "Thus, the pannexin-1-mediated "programmed intracellular ATP decrease" of the apoptotic cell might benefit surrounding healthy cells by preserving environmental glucose."

Results first paragraph – Why is cell-specific information important? This isn't very well explained.

The first sentence should specify whether "between cells" means between cell types or between cells of a given cell type to emphasize the point being made.

Variable time interval range should be described in the text.

“It should be noted that the timing and the speed of the decreases in ATP did vary between cells, emphasizing the advantages of single-cell ATP analysis of apoptotic cells over the conventional cell population-based methods.” Great, but why is that precise information important?

Figure 1—figure supplement 2 should include a negative control consisting of a cohort of cells treated with a control antibody to control for cell death due to imaging conditions (note that during ATP/apoptosis imaging it's not mentioned in the Materials and methods whether the cells are being maintained at 37°C and in 5% CO_2_, however this is noted in the Mg^2+^ level measurements section). Some minimal details of the imaging/acquisition should be provided in the main body of the text.

Figure 1: The data shown in B for a single cell could also be aggregated/averaged and shown for a larger cohort of cells by adjusting the data as in part C. Part C should be added to Figure 1—figure supplement 2 as is the same data presented with an adjusted time scale, not clear how this really adds anything to the story. The images in Part A are much too small to appreciate and spatial information.

Perhaps a more important point is that the images provided in Figure 1A suggest that cell has collapsed between minute 297 and 300, rather than lost ATP, as it appears the FRET signal and TMRE signal are both now (at minute 300) quite spatially restricted but it is not clear that the space where the signal is absent is still "cell". Without a cell membrane marker like WGA (which can be used in live cells), or a DIC image (or similar) it is impossible to ascertain whether or not this is the case.

It seems that Figures 1 and 2 could be collapsed into one figure.

It seems that Figures 3 and 4 should be combined. To more convincingly show that PANX1 is the mediator of the caspase mediated ATP release, the authors should completely delete it using Crispr-Cas9 and then re-express the caspase resistant mutant (and compare with re-expressed WT). There should be at least 3 biological replicates and the traces from the individual biological replicates should be distinguishable from one another.

Figure S6: how many biological replicates here? Reporting on number of cells and number of biological replicates should be consistent throughout the figures.

Figure 5: The logic behind the interpretation of the results of Figure 5 isn't clear, please elaborate on "Strikingly, extracellular AMP and ATP, but not ADP, suppressed the decrease in intracellular ATP levels of apoptotic cells (Figure 5). Thus, it is most likely that the intracellular ATP decrease of apoptotic cells is a result of a reduction in adenosine nucleotide pools inside apoptotic cells, which is caused, at least in part, by release of ATP and AMP from the cells."

Figures 6 and 7: Cytosolic free Mg^2+^ and glucose consumption this also be assayed with Crispr-Cas9 deletion of Panx1, and the compared with wildtype and mutant caspase-defective PANX1 expression.

The conclusions drawn from Figure 7 don't make intuitive sense to me. While the cells aren't consuming the ATP they are still in effect losing ATP by releasing it, so less ATP is available for cellular processes, so they should still "want to consume" glucose.

Finally, given the nature of their studies the authors should take the following papers into consideration and add genetic models to strengthen their investigation of these aspects:

1) Probenecid interferes with renal oxidative metabolism: A potential pitfall in its use as an inhibitor of drug transport https://www.ncbi.nlm.nih.gov/pmc/articles/PMC1572299/

2) Probenecid potentiates MPTP/MPP+ toxicity by interference with cellular energy metabolism. https://www.ncbi.nlm.nih.gov/pubmed/23802648

Reviewer #2:

The major novelty of this paper is the generation of new probes to study intracellular ATP, and using this probe the authors address the involvement of Pannexin-1 in decreasing intracellular ATP (ATPi) during apoptotic cell death via FRET-based single cell imaging. Additionally, albeit in less detail, the authors begin to show that apoptotic cells can use glycolysis during the cell death process but the loss of intracellular ATP impedes this.

Overall usage of the probe to track single cells and their intracellular ATP and some of the controls are well done and the findings are backed up with experimental evidence. However, as the authors acknowledge and based this work on, it is known that ATP is released via pannexin channels during apoptosis and this occurs via caspase-mediated cleavage of the channel. What is potentially novel is the possible differential involvement of glycolysis and OXPHOS during apoptosis when Panx1 function is blocked – however, the data supporting this claim is weak and will require additional experiments to fully understand the contributions of these metabolic processes. Additionally, the authors do not begin to explore what impact this process has on the neighboring tissue, and what the larger importance of it could be. If the authors can successfully address the major concerns, especially major concern 1, 2, and 3, then this work would potentially be a strong candidate for publication in *eLife* as it adds another level to the understanding of cell death process.

1) Although the authors show intracellular ATP decreases during apoptosis, what is the importance of this process? They speculate that there will be more glucose for surrounding cells (which makes them healthier?) because the apoptotic cells cannot use it as they will have lost their ATPi. However, they never actually provide evidence for this and by mentioning it in the Abstract it is misleading. This needs to be addressed much more in detail.

2) This paper primarily addresses one form of death – anti-Fas + methotrexate treatment of HeLa cells. The authors make much larger claims on what this would mean for the larger process of apoptosis and the glycolysis vs. OXPHOS may be impactful for the tissue, etc. But these are not justified. At the minimum, the authors should use one or two other cell lines and another form of cell death.

3) Lastly, the authors provide some indirect evidence for the function of glycolysis during apoptosis. By inhibiting Panx1 channels, thereby increasing ATPi , the authors show an increase in glucose uptake. However, this does not necessarily indicate increased glycolysis. Can the authors measure lactate release to more directly show that there is increased glycolysis during Panx1 inhibition (or even better via Seahorse Assay)? Interestingly, this may indicate that glycolysis is still occurring during apoptosis even if Panx1 is active although it may be with lower levels of ATPi and therefore to a less extent. This is an important point and needs to be proven.

4) The authors attempt to rule-out the involvement of mitochondrial depolarization as a mechanism for decreased ATPi by treatment of CCCP on live cells and showing that ATPi does not decrease. However, this is not a fair comparison because this does not indicate that it cannot be occurring during apoptosis (or in dying cells). Can the authors more clearly rule out this issue? For example, can the authors induce cell death, but inhibit mitochondrial depolarization, and show that ATPi is still decreasing regardless of mitochondria membrane potential? Does this affect the dynamics of ATPi decrease?

5) The authors state that "the ATP decrease in apoptotic cells is a caspase-3-dependent process", but they reach this conclusion based on timing events (correlational). Can the authors inhibit caspases with zVAD/QVD and show that ATPi levels do not decrease after anti-Fas treatment to directly prove this statement?

6) Can the authors show that a cellular ATPi levels still drop in a cell that endogenously/naturally uses OXPHOS and that this is still dependent on Panx1 activation? In this paper they use a system where glucose has been depleted and this can impact results. Also, does Panx1 inhibition still increase glycolysis in these cells? It would be interesting to know whether these cells attempt to switch their metabolism during cell death.

Reviewer #3:

This study describes a novel on-line optical probe to track the kinetics of intracellular ATP concentration in single HeLa cells during the induction/progression of apoptosis by Fas activation. HeLa are "type II" cells wherein extrinsic apoptotic (i.e., receptor-initiated) activation of the executioner caspase-3 is indirectly mediated by a caspase-8> Bid > mitochondrial permeabilization> caspase-9 cascade. This contrasts with type I cells wherein extrinsic apoptosis involves a direct caspase-8 > caspase-3 cascade. The major findings are that the decrease in intracellular ATP: 1) occurs shortly after collapse of mitochondrial membrane potential and accumulation of active caspase-3; and 2) is mediated by caspase-3 mediated proteolytic activation of plasma membrane pannexin-1 channels as conduits for efflux of ATP and other adenine nucleotides. Importantly, genetic or pharmacological suppression of pannexin-1 function markedly delays the decrease in intracellular ATP even when mitochondrial function or ATP synthesis is inhibited. That proteolytically-gated pannexin-1 channels can be major determinants of intracellular and extracellular adenine nucleotide metabolism/ distribution basically confirms several previous studies with populations of apoptotic cells. However, extending such findings to the single cell level is a significant contribution. One major issue requires attention.

Major concern: In the Figure 7 experiments, the authors use 2-deoxyglucose (2DG) as a "glycolysis inhibitor" to conclude that ongoing ATP synthesis via glycolysis is sufficient to sustain the intracellular ATP levels of apoptotic HeLa cell for several hours when pannexin-1-mediated ATP efflux is blocked by probenecid. There's major problem with this interpretation. 2DG acts an intracellular ATP "sink". Hexokinase readily phosphorylates 2DG, but 2DG-phosphate can't be further converted to fructose-6P. Thus, it's difficult to deconvolute the ATP-consuming action of 2DG from subsequent suppression of glycolytic ATP production. This needs to be addressed and alternative experimental support for the stated conclusion is required. Inhibition of glucose uptake by cytochalasins or other GLUT blockers might be an approach.

Reviewer #4:

Too much of this paper is dedicated to demonstration of the role of PANX1 cleavage in ATP release, and in the resulting depletion of the cellular ATP pool, which are, both, not new. (As the authors write "…it has (already) been reported that over-expression of PNAX1-CR (a dominant negative mutant) significantly suppresses the release of ATP from apoptotic cells").

What the paper should have focused at, are the following two things that are new and potentially important:

a) The more accurate information of the kinetics of ATP-loss during apoptosis, and its relation to the kinetics of other changes, which one can gain by applying the ATP

sensor that the authors devised to monitor the changes on a single-cell level.

b) Novel knowledge of the functional significance of the changes in the cellular ATP level in the course of apoptosis.

Both subjects are addressed, but in a way that is not thorough enough to yield solid conclusions.

With regard to the documentation of the kinetics of ATP loss, the paper claims to present findings that challenge a prior claim that the occurrence of apoptosis requests maintenance, or in fact – elevation, of the ATP cellular levels till a rather late stage in the process (Zamaraeva et al., 2005). However, the quality of the evidence that the current paper presents in support of its claims is harmed by the following shortcomings:

a) The means applied for arrest of PANX1 function have limitations. Knock down is incomplete. Probenecid has unrelated effects. The authors should have knocked out

the PANK1 gene by Crispr/Cas9 instead.

b) The cause for the marked intercellular variations in the effect of PANK1 inhibition on the kinetics of ATP decrease is not clear. The authors did not exclude contribution of variation in the extent of arrest of PANK1 function/expression, which makes it particularly important to use cells in which PANK1 was knocked out.

c) The apparent lag in the decrease of ATP pursuant to caspase activation/mitochondrial potential loss (Figure 1B), which might actually be related to the observations of Zamaraeva et al., and the intercellular variation in the length of this lag, requests further analysis and documentation. Does the length of the lag depend on the mode of cell death induction, on the type of cell, on growth condition? The possible mechanisms for this lag should be discussed.

Particularly in view of the dissimilarity to the study of Zamaraeva et al., where various kinds of cells and various means of death induction have been examined, there is need also to apply the tests of the current study in various cells and with various means of death induction. It is of particular importance to repeat the tests in cells induced to die through the type 1 extrinsic apoptotic pathway – a direct Fas (or TNF) >casp8>casp3>Panx1 cascade independent of perturbed mitochondrial function.

d) The evidence presented for some of the major conclusions in this paper (e.g. Figure 1B) is much too limited, constituting just of data acquired with a single cell.

e) It is not clear to which extent does the apoptotic volume-decrease, such as that seen in Figure 1A, affect the interpretation of the data acquired with the ATP sensor. To

which extent does the decrease of the signal indeed reflect decrease of ATP and not decrease in cell area? A clearer more complete description of the methodology e.g. more details of how the background subtractions were made, should be presented

With regard to the functional significance of ATP loss in apoptosis, the paper should have more thoroughly addressed the following three issues:

a) Apoptosis is largely believed to depend on *maintenance* of high ATP levels in the cells. At which exact stage of the apoptotic process does the maintenance of high ATP in the cells turn, from being requested for the apoptotic process, to become inhibitory to some of its manifestations?

b) Does the observed increase in glucose consumption indeed reflect enhanced glycolysis (just monitoring glucose levels does not suffice to tell that), and is this increase also occurring in other cell types, including ones that normally resort to oxidative phosphorylation, and in cells dying in response to other triggers?

c) Which of the various manifestations of apoptosis depend on the decrease of ATP. The authors suggest that, besides its contribution to arrest of glucose consumption, this decrease accounts for the role of PANX1 in restricting blebbing (Yoon et al., Nature 507: 329) and that it also contributes to DNA fragmentation/condensation through

increase of magnesium. However, they provide no evidence for that. One would expect comprehensive assessment of the impact of arrest of ATP decrease (e.g. by incubating the dying cells in growth medium containing ATP or AMP) on a wide range of functional and mechanistic events that are known to be associated with apoptosis.

---

## [Author Response]

[Editors’ note: the authors resubmitted a revised version of the paper for consideration. What follows is the authors’ response to the first round of review.]

Reviewer #1:Detailed comments:Abstract: I'm not sure this sentence in the Abstract makes sense (nor is it substantiated by the data) – "Thus, the pannexin-1-mediated "programmed intracellular ATP decrease" of the apoptotic cell might benefit surrounding healthy cells by preserving environmental glucose."

We have removed this discussion from the Abstract.

Results first paragraph – Why is cell-specific information important? This isn't very well explained.The first sentence should specify whether "between cells" means between cell types or between cells of a given cell type to emphasize the point being made.

We have changed the sentence as follows: “In general, the progression of apoptosis varies between individual cells, even in the same cell type.”

Variable time interval range should be described in the text.

We have added the time information in the text as follows: “The cytosolic ATP levels in these cells started to decrease after a variable time interval (typically from 3 to 8 hours after apoptotic stimulation, see Figure 1—figure supplement 2).”

“It should be noted that the timing and the speed of the decreases in ATP did vary between cells, emphasizing the advantages of single-cell ATP analysis of apoptotic cells over the conventional cell population-based methods.” Great, but why is that precise information important?

This sentence has been removed because it was considered disruptive to the flow of the story.

Figure 1—figure supplement 2 should include a negative control consisting of a cohort of cells treated with a control antibody to control for cell death due to imaging conditions (note that during ATP/apoptosis imaging it's not mentioned in the Materials and methods whether the cells are being maintained at 37°C and in 5% CO_2_, however this is noted in the Mg^2+^ level measurements section). Some minimal details of the imaging/acquisition should be provided in the main body of the text.

Thank you for the suggestions. We added a negative control experiment (Figure 1—figure supplement 2). The cells did not die without the apoptotic stimulus. During imaging, cells were maintained at 37˚C and in 5% CO_2_. We described the conditions in Materials and methods section.

Figure 1: The data shown in B for a single cell could also be aggregated/averaged and shown for a larger cohort of cells by adjusting the data as in part C. Part C should be added to Figure 1—figure supplement 2 as is the same data presented with an adjusted time scale, not clear how this really adds anything to the story. The images in Part A are much too small to appreciate and spatial information.

The old Figure 1 has been removed because we no longer used TMRE as an indicator of caspase-3 activation in the revised experiments.

Perhaps a more important point is that the images provided in Figure 1A suggest that cell has collapsed between minute 297 and 300, rather than lost ATP, as it appears the FRET signal and TMRE signal are both now (at minute 300) quite spatially restricted but it is not clear that the space where the signal is absent is still "cell". Without a cell membrane marker like WGA (which can be used in live cells), or a DIC image (or similar) it is impossible to ascertain whether or not this is the case.

The fluorescence might have appeared to be localized to a part of the cell because the apoptotic cells shrank. There is no spatial restriction of the fluorescent signals. We used larger fluorescent images of the cells in the revised manuscript to clearly show cell shrinkage.

It seems that Figures 1 and 2 could be collapsed into one figure.

These figures have been removed in the revised manuscript.

It seems that Figures 3 and 4 should be combined. To more convincingly show that PANX1 is the mediator of the caspase mediated ATP release, the authors should completely delete it using Crispr-Cas9 and then re-express the caspase resistant mutant (and compare with re-expressed WT). There should be at least 3 biological replicates and the traces from the individual biological replicates should be distinguishable from one another.

We used PANX1-knockout cell lines, instead of using siRNA and inhibitors, in the revised manuscript. Because there are many panels, we decided not to combine figures. Traces from individual replicates were labeled with bars of different shades.

Figure S6: how many biological replicates here? Reporting on number of cells and number of biological replicates should be consistent throughout the figures.

Number of cells and number of biological replicates were presented throughout the revised manuscript.

Figure 5: The logic behind the interpretation of the results of Figure 5 isn't clear, please elaborate on "Strikingly, extracellular AMP and ATP, but not ADP, suppressed the decrease in intracellular ATP levels of apoptotic cells (Figure 5). Thus, it is most likely that the intracellular ATP decrease of apoptotic cells is a result of a reduction in adenosine nucleotide pools inside apoptotic cells, which is caused, at least in part, by release of ATP and AMP from the cells."

I agree that the explanation in the previous manuscript was ambiguous. We have added the following sentences in the revised manuscript: “If efflux of adenine nucleotides through PANX1 channel causes the decrease in the cytosolic ATP concentrations of apoptotic cells, extracellular adenine nucleotides would suppress the decrease by counteracting the efflux of its cytosolic counterpart.”

Figures 6 and 7: Cytosolic free Mg^2+^ and glucose consumption this also be assayed with Crispr-Cas9 deletion of Panx1, and the compared with wildtype and mutant caspase-defective PANX1 expression.

We re-measured glucose consumption and free Mg^2+^ dynamics using PANX1-knockout cells.

The conclusions drawn from Figure 7 don't make intuitive sense to me. While the cells aren't consuming the ATP they are still in effect losing ATP by releasing it, so less ATP is available for cellular processes, so they should still "want to consume" glucose.

In general, activity of enzymes, including ATPases, highly depends on substrate concentration. Although I agree that some ATPases with very low *K*_m_ will retain their activity even at low ATP levels, the decreased intracellular ATP levels will reduce the net ATPase activity of the cell. We added the following sentence in the revised manuscript: “It is also likely that the reduction of ATP during apoptosis leads to the decrease in the activities of other ATPases because ATPase activity depends on the concentration of ATP.”

Finally, given the nature of their studies the authors should take the following papers into consideration and add genetic models to strengthen their investigation of these aspects:1) Probenecid interferes with renal oxidative metabolism: A potential pitfall in its use as an inhibitor of drug transport https://www.ncbi.nlm.nih.gov/pmc/articles/PMC1572299/2) Probenecid potentiates MPTP/MPP+ toxicity by interference with cellular energy metabolism. https://www.ncbi.nlm.nih.gov/pubmed/23802648

Thank you for raising the concern. We decided to remove the data that used probenecid.

Reviewer #2:[…]1) Although the authors show intracellular ATP decreases during apoptosis, what is the importance of this process? They speculate that there will be more glucose for surrounding cells (which makes them healthier?) because the apoptotic cells cannot use it as they will have lost their ATPi. However, they never actually provide evidence for this and by mentioning it in the Abstract it is misleading. This needs to be addressed much more in detail.

Metabolism is one of the major biological activity of living systems. The metabolic activity of cells is an indicator of "the state of being alive". In other words, in order for a cell to "die a complete death", metabolism must stop. However, it has not been well understood how metabolism of dead cells ceases. Our results showed that PANX1 plays a major role in stopping glycolysis, a central part of metabolism, of apoptotic cells, through facilitating ATP loss. This is the biological importance of ATP depletion in apoptosis that we want to emphasize in the revised work. We discuss this statement in the revised manuscript. We agree that the model that PANX1 stops the waste of glucose by apoptotic cells to benefit surrounding cells is a discussion/speculation. We have deleted this point from Abstract.

2) This paper primarily addresses one form of death – anti-Fas + methotrexate treatment of HeLa cells. The authors make much larger claims on what this would mean for the larger process of apoptosis and the glycolysis vs. OXPHOS may be impactful for the tissue, etc. But these are not justified. At the minimum, the authors should use one or two other cell lines and another form of cell death.

In the revised manuscript, we analyzed staurospoine-induced apoptosis (Figure 2E-F, 4DE), as well as anti-FAS + cycloheximide-induced apoptosis for HeLa cells. In addition,

TRAIL-induced apoptosis was analyzed for SW480 cells (Figure 2—figure supplement

1).

3) Lastly, the authors provide some indirect evidence for the function of glycolysis during apoptosis. By inhibiting Panx1 channels, thereby increasing ATPi , the authors show an increase in glucose uptake. However, this does not necessarily indicate increased glycolysis. Can the authors measure lactate release to more directly show that there is increased glycolysis during Panx1 inhibition (or even better via Seahorse Assay)? Interestingly, this may indicate that glycolysis is still occurring during apoptosis even if Panx1 is active although it may be with lower levels of ATPi and therefore to a less extent. This is an important point and needs to be proven.

Thank you for the suggestion. Lactate release was also measured in the revised manuscript. Consistent with glucose consumption data, lactate release did not cease in PANX1-KO cells. To further support that existence of glycolytic activity, we showed that lactate dehydrogenase inhibitor sodium oxamate also reduces intracellular ATP, like hexokinase inhibitor 2-deoxyglucose does. Please note that we did not show that glycolysis is increased, just showed that glycolysis is retained at substantial levels in apoptotic cells.

4) The authors attempt to rule-out the involvement of mitochondrial depolarization as a mechanism for decreased ATPi by treatment of CCCP on live cells and showing that ATPi does not decrease. However, this is not a fair comparison because this does not indicate that it cannot be occurring during apoptosis (or in dying cells). Can the authors more clearly rule out this issue? For example, can the authors induce cell death, but inhibit mitochondrial depolarization, and show that ATPi is still decreasing regardless of mitochondria membrane potential? Does this affect the dynamics of ATPi decrease?

We have no idea to perform the suggested experiment. The relevant text and data have been removed in the revised version for clarity. We do not think that the removal of this result affects the major conclusion of this study.

5) The authors state that "the ATP decrease in apoptotic cells is a caspase-3-dependent process", but they reach this conclusion based on timing events (correlational). Can the authors inhibit caspases with zVAD/QVD and show that ATPi levels do not decrease after anti-Fas treatment to directly prove this statement?

We have confirmed that treatment with zVAD inhibit ATPi decrease after anti-FAS stimulation (Figure 1A).

6) Can the authors show that a cellular ATPi levels still drop in a cell that endogenously/naturally uses OXPHOS and that this is still dependent on Panx1 activation? In this paper they use a system where glucose has been depleted and this can impact results. Also, does Panx1 inhibition still increase glycolysis in these cells? It would be interesting to know whether these cells attempt to switch their metabolism during cell death.

As far as we know, cultured mammalian cells that are generally used for apoptosis research predominantly use glycolysis for ATP synthesis rather than OXPHOS. Therefore, it is difficult to conduct experiments to solve this question at present.

Reviewer #3:[…]Major concern: In the Figure 7 experiments, the authors use 2-deoxyglucose (2DG) as a "glycolysis inhibitor" to conclude that ongoing ATP synthesis via glycolysis is sufficient to sustain the intracellular ATP levels of apoptotic HeLa cell for several hours when pannexin-1-mediated ATP efflux is blocked by probenecid. There's major problem with this interpretation. 2DG acts an intracellular ATP "sink". Hexokinase readily phosphorylates 2DG, but 2DG-phosphate can't be further converted to fructose-6P. Thus, it's difficult to deconvolute the ATP-consuming action of 2DG from subsequent suppression of glycolytic ATP production. This needs to be addressed and alternative experimental support for the stated conclusion is required. Inhibition of glucose uptake by cytochalasins or other GLUT blockers might be an approach.

We agree that 2DG can potentially act as an ATP sink. In the revised manuscript, we also used sodium oxamate, an inhibitor for lactate dehydrogenase. Supply of NAD+, which is required for the reaction of GAPDH, is inhibited by this chemical. Consistent with our model, the addition of sodium oxamate to the apoptotic PANX1-KO cells rapidly reduced the intracellular ATP levels (Figure 7C).

Reviewer #4:Too much of this paper is dedicated to demonstration of the role of PANX1 cleavage in ATP release, and in the resulting depletion of the cellular ATP pool, which are, both, not new. (As the authors write "…it has (already) been reported that over-expression of PNAX1-CR (a dominant negative mutant) significantly suppresses the release of ATP from apoptotic cells").

The reviewer might think that nucleotide release is synonymous with depletion of intracellular ATP concentration, but we do not think that the idea is correct. For example, astrocytes and vascular endothelial cells are known to release ATP frequently, but the ATP release is unlikely to deplete ATP in these cells. Most of previous PANX1-apoptosis works have focused on nucleotide release. Therefore, we disagree with the reviewer's view that there is no scientific novelty in examining in detail whether the mechanism of intracellular ATP depletion in apoptosis is related to the mechanism of nucleotide release.

What the paper should have focused at, are the following two things that are new and potentially important:a) The more accurate information of the kinetics of ATP-loss during apoptosis, and its relation to the kinetics of other changes, which one can gain by applying the ATPsensor that the authors devised to monitor the changes on a single-cell level.b) Novel knowledge of the functional significance of the changes in the cellular ATP level in the course of apoptosis.Both subjects are addressed, but in a way that is not thorough enough to yield solid conclusions.

Thank you for the positive comments on our work. We believe that the revised work has strengthen these conclusions.

With regard to the documentation of the kinetics of ATP loss, the paper claims to present findings that challenge a prior claim that the occurrence of apoptosis requests maintenance, or in fact – elevation, of the ATP cellular levels till a rather late stage in the process (Zamaraeva et al., 2005). However, the quality of the evidence that the current paper presents in support of its claims is harmed by the following shortcomings:a) The means applied for arrest of PANX1 function have limitations. Knock down is incomplete. Probenecid has unrelated effects. The authors should have knocked outthe PANK1 gene by Crispr/Cas9 instead.b) The cause for the marked intercellular variations in the effect of PANK1 inhibition on the kinetics of ATP decrease is not clear. The authors did not exclude contribution of variation in the extent of arrest of PANK1 function/expression, which makes it particularly important to use cells in which PANK1 was knocked out.c) The apparent lag in the decrease of ATP pursuant to caspase activation/mitochondrial potential loss (Figure 1B), which might actually be related to the observations of Zamaraeva et al., and the intercellular variation in the length of this lag, requests further analysis and documentation. Does the length of the lag depend on the mode of cell death induction, on the type of cell, on growth condition? The possible mechanisms for this lag should be discussed.Particularly in view of the dissimilarity to the study of Zamaraeva et al., where various kinds of cells and various means of death induction have been examined, there is need also to apply the tests of the current study in various cells and with various means of death induction. It is of particular importance to repeat the tests in cells induced to die through the type 1 extrinsic apoptotic pathway – a direct Fas (or TNF) >casp8>casp3>Panx1 cascade independent of perturbed mitochondrial function.

Because our results indicate that cytosolic ATP is maintained until caspase-3 is activated (Figure 1C-D), we think that our results do not fully challenge the previous claim that the occurrence of apoptosis requests maintenance of ATP. However, we did not observe an apparent increase in cytosolic ATP just after induction of apoptosis, unlike the observation by Zamaraeva and colleagues. The cause of this discrepancy is unknown at present. According to the reviewer’s suggestion, we have performed the same experiments again using PANX1 knockout cells and using different types of apoptotic stimulations. Again, we observed the short lag in ATP reduction and the kinetic variations in cytosolic ATP reduction after caspase-3 activation. The possible causes for the lag and the variations were discussed in the revised manuscript as described below.

“Notably, the intracellular ATP levels of PANX1-KO cells were almost unchanged in the first 30-60 min after the activation of caspase-3, followed by a gradual ATP decline (Figure 2C-F, Figure 2—figure supplement 1). The lag in the cytosolic ATP decrease observed for PANX1-KO cells might be partially relevant to the previous observation by Zamaraeva (Zamaraeva et al., 2005), which suggested the enhancement of cytosolic ATP level after apoptotic stimulation.”

In addition, we performed the experiments using SW480 cells (Figure 2—figure supplement 1), which is known to die through type 1 cell death pathway when stimulated with TRAIL (Özören and El-Deiry 2002, Neoplasia).

d) The evidence presented for some of the major conclusions in this paper (e.g. Figure 1B) is much too limited, constituting just of data acquired with a single cell.

We disagree with this assertion. We think that the power of single-cell analysis has led to these results. In addition, the single-cell ATP imaging technology, which is the basis of this research, has been proven to be practical by number of researchers over the past decade.

e) It is not clear to which extent does the apoptotic volume-decrease, such as that seen in Figure 1A, affect the interpretation of the data acquired with the ATP sensor. Towhich extent does the decrease of the signal indeed reflect decrease of ATP and not decrease in cell area? A clearer more complete description of the methodology e.g. more details of how the background subtractions were made, should be presented

Since we monitor the ratio of YFP fluorescence and CFP fluorescence as a signal, any increase or decrease in fluorescence intensity due to cell shrinkage is offset. That is, this method is much more robust against morphological changes in cells than methods such as obtaining a single fluorescence wavelength. We added the following sentence. “It should be noted that any increase or decrease in fluorescence intensity due to cell morphological change was offset because we monitored the ratios of fluorescence intensities of an acceptor and a donor of the FRET biosensors.”

Details of background subtractions and ratio calculation were described in Materials and methods.

With regard to the functional significance of ATP loss in apoptosis, the paper should have more thoroughly addressed the following three issues:a) Apoptosis is largely believed to depend on maintenance of high ATP levels in the cells. At which exact stage of the apoptotic process does the maintenance of high ATP in the cells turn, from being requested for the apoptotic process, to become inhibitory to some of its manifestations?

Previous studies have shown that the pre-reduction of ATP inhibits the activation of caspase-3 (Zamaraeva et al., 2005) and switches cell death form from apoptosis to necrosis (Eguchi, Shimizu and Tsujimoto, 1997; Leist et al., 1997). It is also known that dATP/ATP is also important for the formation of apoptosomes (Li et al., 1997; Hu et al., 1997). Our results demonstrated that cytosolic ATP is retained at high level until caspase-3 is activated. Thus, it is very likely that maintenance of high ATP is required until the activation of caspases. We added the above points into the revised manuscript (Discussion paragraph one). We found that PANX1-KO did not affect phosphatidylserine exposure, indicating that ATP loss is not critical to this process (Figure 6B). Apoptotic PANX1-KO cells did not immediately die a complete death because they did not completely stop glycolytic metabolism (Figure 7B). Also, apoptotic PANX1-KO cells continued to waste environmental glucose for more than 32 hours, which potentially causes starvation of the surrounding cells (Figure 7F-G and Figure 8).

b) Does the observed increase in glucose consumption indeed reflect enhanced glycolysis (just monitoring glucose levels does not suffice to tell that), and is this increase also occurring in other cell types, including ones that normally resort to oxidative phosphorylation, and in cells dying in response to other triggers?

Glycolytic activity of apoptotic cell is *not* enhanced, as evident from Figure 6E-G: Although apoptotic PANX1-KO cells retain glycolytic activity, the glucose consumption rate and the lactate release rate are slower than the wild-type cells.

c) Which of the various manifestations of apoptosis depend on the decrease of ATP. The authors suggest that, besides its contribution to arrest of glucose consumption, this decrease accounts for the role of PANX1 in restricting blebbing (Yoon et al., Nature 507: 329) and that it also contributes to DNA fragmentation/condensation throughincrease of magnesium. However, they provide no evidence for that. One would expect comprehensive assessment of the impact of arrest of ATP decrease (e.g. by incubating the dying cells in growth medium containing ATP or AMP) on a wide range of functional and mechanistic events that are known to be associated with apoptosis.

We added new videos as supplementary materials, which clearly shows that an apoptotic PANX1-KO cell immediately stop blebbing after the forced decrease in intracellular ATP concentration by 2DG treatment. We think that this video strongly supports the idea that PANX1-enhanced decrease in intracellular ATP restricts excessive blebbing of apoptotic cells.